# Electrochemical Biosensors in Food Safety: Challenges and Perspectives

**DOI:** 10.3390/molecules26102940

**Published:** 2021-05-15

**Authors:** Antonella Curulli

**Affiliations:** Istituto per lo Studio dei Materiali Nanostrutturati (ISMN) CNR, Via del Castro Laurenziano 7, 00161 Roma, Italy; antonella.curulli@cnr.it

**Keywords:** food, safety, electrochemical biosensors, bacteria, toxins, pesticides, antibiotics, contaminants

## Abstract

Safety and quality are key issues for the food industry. Consequently, there is growing demand to preserve the food chain and products against substances toxic, harmful to human health, such as contaminants, allergens, toxins, or pathogens. For this reason, it is mandatory to develop highly sensitive, reliable, rapid, and cost-effective sensing systems/devices, such as electrochemical sensors/biosensors. Generally, conventional techniques are limited by long analyses, expensive and complex procedures, and skilled personnel. Therefore, developing performant electrochemical biosensors can significantly support the screening of food chains and products. Here, we report some of the recent developments in this area and analyze the contributions produced by electrochemical biosensors in food screening and their challenges.

## 1. Introduction

Food safety is an important critical issue for the modern food industry. Contaminants, bacteria, toxins, etc., can enter the food chain during the production of different steps. For example, they can accumulate in food during storage and/or are produced in the food by reaction with chemical compounds [1].

A preventative approach to food safety is the hazard analysis critical control point (HACCP), which attempts to avoid the contamination of unwanted substances into the food chain [2,3]. On the other hand, some rigid guidelines are defined by the regulatory agencies, such as United States Food and Drug Administration (USFDA) and the European Food Safety Authority (EFSA). These guidelines indicate the maximum levels for contaminants in foods to preserve consumer health [4,5].

Food analysis is carried out at the end of the production process using conventional techniques, such as chromatography, mass spectrometry, ultraviolet detection, or fluorescence techniques either individually or combined with other separation techniques [6,7]. These traditional approaches have several limitations. First, since the analysis is performed at the end of the process, contaminated products can pass through the entire production chain or even be placed on the market before contamination is noticed. Second, these analysis methods are laborious and complex, expensive, time-consuming, require large sample volumes and skilled personnel [8].

In this context, biosensors can offer a possible alternative to allow the screening of food samples before the end of the production process [8]. Furthermore, biosensors also provide rapid and on-site monitoring and real-time information about the production process [9]. Among various biosensors, electrochemical biosensors have been widely used due to their well-understood biointeraction mechanisms and detection process [10]. Electrochemical biosensors can represent smart detection tools for food commodities as part of an accurate, sensitive, specific, and rapid analysis system [11,12].

In this review, we consider recently developed electrochemical biosensors applied for food analysis and safety. We illustrate recent advances in biosensing technologies and evaluate their related weaknesses and drawbacks. We include some future ideas and challenges that electrochemical biosensors must overcome to be new and smart tools for food analysis and safety.

## 2. Electrochemical Biosensors

A biosensor is an analytical device used to determine the amount of a molecule in a sample. Generally, it is characterized by a bioreceptor (enzyme, whole-cell, antibody, aptamer, nucleic acid) connected to a suitable transducer. The specific interaction between the target molecule and the biocomponent generates a physicochemical or biological signal, converted into a measurable property by the transducer. The choice of the bioreceptor and the transducer depends on the sample’s characteristics and the type of measurable property being considered. The bioreceptor represents the biosensor key element, responding only to a particular analyte and not to the interferences eventually present in the sample under analysis [13,14,15].

### 2.1. Electrochemical Biosensors Classification

Biosensors can be classified by type of recognition element or type of signal transduction [16], as indicated in Figure 1.

In this review, we focus on electrochemical biosensors. Very interesting and recent reviews have illustrated the characteristics and performances of the other biosensors with different transducer systems, such as optical, piezoelectric, calorimetric [1,3,17,18,19].

Among the different typologies of biosensors, electrochemical ones combine the sensitivity, as indicated by low detection limits, of electrochemical transducers with the high specificity of biorecognition processes [10]. These devices contain a biological recognition element, like the other biosensors (enzymes, proteins, antibodies, nucleic acids, cells, tissues, or receptors), reacting specifically with the target analyte and producing an electrical signal related to the concentration of the analyte. A schematic representation of an electrochemical biosensor is shown in Figure 2.

Electrochemical biosensors can be divided into two main categories based on the nature of the biological recognition process, i.e., biocatalytic devices and affinity biosensors [2,20,21]. Biocatalytic devices incorporate enzymes, whole cells or tissue slices that recognize the target analyte and produce electroactive species.

Affinity biosensors are based on a selective binding interaction between the analyte and a biological component, such as an antibody, nucleic acid, or receptor. Immunosensors and DNA hybridization biosensors with electrochemical detection are considered examples of affinity sensors.

In the first case, the recognition element can be characterized by enzymes, whole-cells (bacteria, fungi, cells, yeast), cells organelle and plant or animal tissue slices; the catalytic sensors have the most consolidated tradition in the field of biosensors: historically, glucose biosensors are the most cited examples, including a wide commercial success and diffusion [20].

If the recognition key is an enzyme, it is the most critical component of the biosensor since it provides the selectivity for the sensor and catalyzes forming the electroactive product to be detected. The electroactive product or, alternatively, the disappearance of the redox-active reactant in an enzyme-catalyzed reaction can be monitored by the electrode using an electrochemical technique. The activity of the immobilized enzyme depends on solution parameters and electrode design.

Some biocatalytic sensors can use as recognition element cellular materials (whole-cells or tissue slices). These biocatalytic electrodes act similarly to the conventional enzyme electrodes since enzymes present in the tissue or cell can produce or consume electrochemically detectable species. Whole cells and tissue slices are sometimes a better source of enzymatic activity than isolated enzymes as some enzymes are expensive or not commercially available as purified enzymes. In addition, many isolated enzymes have limited stability and lifetime compared to enzymes in their native environment. However, the sensor response may be slower for these sensors because of a more difficult diffusion of the substrate through a thick tissue material.

Unfortunately, many biochemical analytes of interest are not suitable to be detected by enzyme electrodes because of the lack of selectivity for the analyte or the analyte not being commonly found in living/biological systems. For these reasons, affinity biosensors are considered a good option.

Considering the affinity-based biosensors, the biomolecule can be represented by chemoreceptors, antibodies, nucleic acids, and they provide selective interactions with the analyte [2,21].

Affinity sensors use the selective and strong binding of biomolecules, such as antibodies (Ab), membrane receptors, or oligonucleotides, with a target analyte to produce a measurable electrical signal. The molecular recognition is mainly determined by the complementary size and shape of the binding site for the analyte of interest. The high affinity and specificity of the biomolecule for its ligand make these sensors very sensitive and selective. The binding process, such as DNA hybridization or antibody–antigen (Ab–Ag) complexation, is governed by thermodynamic considerations and rules.

Immunosensors are Ab-based affinity biosensors where the detection of an analyte, an antigen or hapten, is induced by its binding to a region/site of an Ab.

The electrochemical transducer reacts to the binding event and converts the electrical response to an easily handled output. Complementary regions of the Ab bind to an Ag, used for producing the antibodies in a host organism with high specificity and affinity. Such polyclonal Abs are heterogeneous concerning their binding domain and may be refined by a selection process to yield monoclonal Abs-MAbs. All of the members of a particular MAb clone are identical. Abs and MAbs can be developed for a wide range of substances.

Immunosensors are well-known for their extremely low detection limits. For this reason, immunosensors can be used to detect trace levels (ppb, ppt) of bacteria, viruses, drugs, hormones, pesticides, and numerous other chemical compounds. Examples of immunosensor applications, including monitoring food safety detecting toxins, bacteria allergens, contaminants, such as pesticides, endocrine disruptors, and drugs, are included in this review (see Section 3).

Nucleic acids have been commonly used as the biorecognition element in affinity sensors. Biorecognition using DNA or RNA nucleic acid fragments is based on either complementary base-pairing between the sensor nucleic acid sequence and the analyte of interest or generating nucleic acid structures, known as aptamers, recognizing and binding to three-dimensional surfaces, such as those of proteins. Nucleic acids are now becoming of greater importance as the biorecognition agent in sensors since a recent rapid expansion in knowledge of their structure and how to manipulate them. The corresponding affinity probes are commonly indicated as aptasensors and nowadays are widely applied in food analysis, as shown in Section 3.

As concerns the electrochemical biosensors, measurements of signals from biological samples are generally linked to an electrochemical reaction involving a bio element electrochemically active. Usually, biological reactions can generate either a change in conductance or impedance, a measurable current, or charge accumulation, measured by conductometric, potentiometric, or amperometric techniques [22,23]. Investigated reactions are normally detected near the electrode surface, and the detection techniques are generally chosen considering the electrochemical properties of the electrode surface. Electrochemical techniques involve reference, auxiliary, and working/sensing electrodes.

The working/sensing electrode acts as a transduction element, whereas a counter electrode establishes a connection between the solution and the sensing electrode surface [23].

Electrochemical techniques have been considered useful tools for food safety analysis. They are cheap, portable, easy to handle, and fast. Thus they can be preferred to the other analytical techniques. For more details about theories underlying the different electrochemical approaches used in the biosensing area, several books and reviews in the literature are well-known [22,24,25,26,27,28,29].

Classified on the transduction principle and then on the corresponding electrochemical technique, the electrochemical biosensors are categorized as (a) Potentiometric, (b) amperometric, (c) impedimetric, (d) conductometric, and (e) voltammetric, as shown in Figure 3.

Potentiometric biosensors used ion-selective electrodes for measuring the change in potential at the surface of the working electrode upon specific analyte–bioreceptor interaction. These biosensors are widely used for different bioanalytes, such as glucose, triglycerides, and pesticides. However, potentiometric transducers generally lack sensitivities when compared to amperometric transducers.

Potentiometry (PM) measures the potential of a solution between two electrodes is used in electroanalytical chemistry to measure the electrochemical potential of charged species. However, a highly stable and accurate reference electrode is required, limiting applying PM in bioanalysis.

In an amperometric biosensor, the current produced at the working electrode through the conversion of electroactive moieties is measured when a certain potential is applied concerning the reference electrode. The current so produced is directly related to oxidation or reduction of the analyte species after its specific interaction with the bioreceptor in proportion to the concentration of target components. The amperometric biosensors are relatively simple and easy to use while also offering relatively high sensitivities.

Compared with the potentiometric biosensors, this method allows sensitive, fast, precise, and linear response, resulting in more suitable for mass production. However, poor selectivity and interferences from other electroactive substances are the disadvantages of these sensors.

Generally, the widest used amperometric techniques are constant potential amperometry and chronoamperometry.

Constant potential amperometry (CPA) is an electrochemical technique in which the current is measured, while the potential difference between the sensing and reference electrodes is held at a constant value sufficient to oxidize or reduce the analyte. This potential value is generally evaluated from the CV or LSV experiment

Chronoamperometry (CA) is a potentiostatic technique, where the current is recorded as a time function, and it is useful to determine the concentration of the analyte once its identity is known using other techniques, such as CV, chromatography and/or other separation techniques.

Voltammetric biosensors detect analytes by measuring the current during the controlled variation of the applied potential. Advantages of these sensors include highly sensitive measurements and possible simultaneous detection of multiple analytes.

Voltammetry is an analytical technique in which the current is measured under a potential sweep. In a voltammogram, the intensity of a peak is directly proportional to the concentration of the analyte, while the position of the peak maximum depends on the chemical species involved in the redox reactions. The type of voltammetry depends on the potential control. Differential pulse voltammetry (DPV), cyclic voltammetry (CV) and square wave voltammetry (SWV) are the most commonly used to detect pathogenic bacteria in food. Cyclic voltammetry is also often used to characterize the surface and the various functionalization steps in all types of biosensors.

Conductometric biosensors quantify the change in the conductance between the pair of electrodes due to an electrochemical reaction (change in conductivity properties of the analyte). Conductometric and impedimetric biosensors are usually used to monitor metabolic processes in living biological systems.

Impedimetric biosensors measure the electrical impedance produced at the electrode/electrolyte interface when a small sinusoidal excitation signal is applied.

Electrochemical impedance spectroscopy (EIS) is an effective technique for detecting the interaction between bioreceptor immobilized on an electrode surface and the analyte by testing the electrode/electrolyte interface and following the change in the impedance of the electrode/solution interface.

In general, more comprehensive and complete information about the biosensing system can be obtained from EIS when compared to that one obtained from the more usual voltammetric techniques. EIS can distinguish between two or more electrochemical reactions occurring simultaneously and can identify diffusion-limited reactions. The impedimetric technique makes biosensors label-free, highly sensitive, and miniaturized. In these biosensors, the interaction of the analyte–bioreceptor is correlated with the change in impedance (Z) across the surface of the working electrode. The Z values are studied to determine the change in frequency concerning time. The impedance data are generally represented in the form of a Nyquist plot, in which the real component of impedance (Z’) and the imaginary component of impedance (Z”) are plotted on the *x-* and *y*-axes, respectively. Note that the impedance is made of two major parts, i.e., resistance (*R*) and capacitance (*C*). At high frequency, the solution resistance accounts for the impedance. In contrast, at low-frequency regions, the charge transfer resistance (*R*ct) or the resistance to the flow of electrons is the source of impedance.

### 2.2. Biorecognition Keys: Bioreceptors

The bioreceptors must interact specifically with the target analyte to generate a measurable signal by the transducer. As mentioned above, when the classification of biosensors is reported, they are enzymes, antibodies, nucleic acids, and aptamers. In addition, other recognition elements, such as synthetic aptamers, DNA, proteins, and viruses, have improved the selectivity of sensors for food analysis. Further, developing innovative bioconjugation approaches for stable immobilization of biomolecules on the electrode surface has enhanced the stability of biosensors. The introduction of nanomaterials in biosensors has impacted the sensitivity of sensors with a high surface area to volume ratio to strengthen the loading capacity of biomolecules relative to the biosensors assembled without nanomaterials.

### 2.3. Sensing Materials and Electrodes

The transducer is the most important component of a biosensor because it directly affects the sensor performances, such as sensitivity and response time. The chemical reaction occurring in the sensing layer near the electrode surface is transformed into an electrochemical signal. The rate and the quality of signal production are directly related to the surface properties of the electrode, the rate of electron transfer, and mass transfer. Thus, selecting electrode material highly affects the rate of electron transfer in electrochemical biosensors. Various types of electrodes used in electrochemical biosensors are reported below.

The peculiar properties of gold (e.g., biocompatibility, stability, and conductivity) have promoted its use as electrodes in electrochemistry. The gold electrode sensitivity and functionality can be enhanced by modifying their surface, introducing suitable molecules and polymers.

For example, long-chain organic compounds, such as thiol, have been employed to modify gold surfaces using self-assembling monolayers (SAM), which can be used as anchoring/immobilizing platforms of enzymes or specific bioreceptors. Such modified electrodes have been applied preferentially in several examples of biosensors.

In addition, gold nanomaterials were employed in the electrochemical biosensing area not only for their high conductivity, their compatibility but also for their high surface to volume ratio [30,31].

Carbon has been recognized as one of the most common electrode materials used in electrochemical biosensing. The most common forms of carbon used as electrode materials are carbon paste, glassy carbon, carbon nanotubes, and graphene electrodes. All these carbon materials are cheaper than noble metals.

Carbon paste is made of graphite powder and an organic binder, immiscible with water and is useful in insulating graphite from aqueous solutions. This carbon-based electrode presents several advantages, such as low cost, low background current, regenerability, and various operating potentials. Moreover, different compounds can be easily incorporated into the carbon paste for bioanalytical applications.

Similarly, glassy carbon electrodes have also been employed for electrochemical biosensors using ad hoc modifications. However, apart from their high cost, glassy carbon electrodes need an accurate pretreatment procedure, constraining their use in many electrochemical applications.

Carbon nanotubes (CNTs) present several properties associated with their structure, functionality, morphology, and flexibility and can be classified as single-walled nanotubes (SWNTs), double-walled nanotubes (DWNTs), and multi-walled nanotubes (MWNTs) depending on the number of graphite layers. [32] Functionalized CNTs have been used in several application fields. The chemical functionalities for their application in biosensing can easily be designed and tuned through tubular structure modification.

Graphene is one of the most applied nanomaterials in the sensing area. Different graphene-based materials have been produced (e.g., electrochemically and chemically modified graphene) using many procedures [33]. Graphene shows properties, such as high conductivity, speeding up electron transfer, and a large surface area, very similar indeed to the corresponding properties of CNTS, so it is considered a good candidate for assembling sensors to determine several target molecules.

Graphene oxide (GO) is hydrophilic and can be dispersed in water solution because of the presence of hydrophilic functional groups (OH, COOH and epoxides). On the other hand, GO has a lower conductivity than graphene, so reduced GO (rGO) is more employed as the electrode modifier in the electrochemical biosensing area [33].

Finally, nonconventional sensing platforms, such as paper and/or screen-printed electrodes (SPE), frequently modified with different nanomaterials and/or nanostructures, are employed in assembling electrochemical biosensors.

Screen-printing technology offers several advantages for assembling electrochemical biosensors, including a wide range of geometries, mass production, disposability, and portability. [34] These properties are very important for commercializing biosensors.

Recent developments in the fabrication of screen-printing electrodes (SPEs) were the topic not only of numerous original research papers but also of interesting reviews, [35,36] analyzing selecting support material, ink composition, and methods of surface modification or functionalization. Finally, in all the reviews mentioned above, methods of obtaining well-defined geometries and microelectrode arrays are discussed and compared for assembling smart electrochemical biosensors.

## 3. Application of Electrochemical Biosensors in Food Analysis

This review focused on the electrochemical biosensors as smart analytical tools to detect some of the most important bacteria, toxins, pesticides, antibiotics, and contaminants in foods.

### 3.1. Toxins

Toxins are present in a natural environment. They are produced by microbes and algae. According to their origin, toxins are commonly classified into bacterial toxins, fungal toxins, and algal toxins. [37] Toxins contamination is unforeseeable and inevitable. In fact, it can take place during the food production chain, including processing, transport, and storage, so causing severe economic losses and public health problems. Based on the survey from the World Health Organization (WHO), humans are exposed to toxins through the ingestion of contaminated foods, causing severe poisoning [38,39].

Herein, this review investigates the state-of-art of the electrochemical biosensors to detect toxins with a particular focus on several typical toxins, such as shellfish toxins, algae toxins, and mycotoxins, and Table 1 summarizes the analytical characteristics of recent electrochemical biosensors for toxins reported in the review.

#### 3.1.1. Shellfish Toxins

Most shellfish toxins are small molecules, usually produced by toxic algae and accumulated in shellfish [40].

Wu et al. reported an overview of the different and widely used approaches in biosensing for shellfish toxins detection [41], emphasizing the importance of electrochemical biosensors and of impedimetric ones.

Herein, some interesting examples of innovative approaches to determining shellfish toxins, such as saxitoxin (STX), domoic acid (DA), and okadaic acid (OA), are reported.

The European Safety Authority (EFSA) [42] indicated provisional acute reference doses (ARfDs) for the OA, STX, and DA toxins 0.33 mg/kg, 0.7 mg/kg and 100 mg/kg, respectively.

The acute reference dose is the estimate of the amount of substance in food, normally expressed on a body–weight basis (mg/kg or μg/kg of body weight), that can be ingested in a period of 24 h or less without appreciable health risk to the consumer based on all known facts at the time of evaluation [42].

Wang and coworkers reported a label-free electrochemical aptasensor assembled with nanotetrahedron and aptamer triplex for sensitive detection of saxitoxin [43].

The aptamer technology, DNA nanotetrahedron, DNA triplex, and electrochemistry were combined for the first time to construct a label-free electrochemical aptasensor for the sensitive detection of small molecules.

A typical small molecule, saxitoxin, was chosen as a model target, considering its low molecular weight and high toxicity. STX is one of the major toxins of paralytic shellfish poison (PSP) and can cause shock, asphyxia and even death to fisheries and humans [44].

Some concepts such as aptasensors, DNA nanotetrahedron must be introduced.

Aptamers are binding oligonucleotides molecules generated by systematic evolution of ligands by exponential enrichment (SELEX), showing the high affinity and high selectivity towards their specific molecular targets. Aptamers have attracted particular attention, especially in the research areas targeting small molecules, owing to aptamers’ advantages, such as in vitro selection, rapid chemical synthesis, and easy chemical modification [45]. Many aptamers showing high affinity and selectivity vs. small molecules have been selected, such as aptamers towards marine toxins [46], mycotoxins [47], pesticides [48], etc. Various aptasensors were developed in the past decades [49]. The electrochemical aptasensors can involve easy handling and rapid response [50,51], allowing a direct capture of the molecule target. The applicability of electrochemical aptasensors towards small molecules is constantly evolving and developing, and it is still under investigation [52].

To overcome some limitations of electrochemical aptasensors for small molecules, the aptamer and DNA triplex were combined and assembled with the nanotetrahedron to form one DNA structure, followed by immobilization on the surface of screen-printed electrodes [43].

Nanotetrahedron (NTH), a rigid DNA nanostructure assembled by four single-stranded monomers, is a spacer for the oriented immobilization of DNAs on surfaces. With the assistance of nanotetrahedron, the absorption of the immobilized DNAs was eliminated, and the target’s access to the immobilized DNAs was facilitated.

DNA triplex is formed by a third DNA strand composed of homopurine or homopyrimidine bonded to a DNA duplex.

The nanotetrahedron assisted the oriented immobilization of the aptamer triplex on the surface of screen-printed electrodes, protecting the aptamer triplex from absorption and assisting the aptamer to show full accessibility to STX. The developed aptasensor provided high sensitivity with a LOD of 0.92 nM and showed good applicability to detect STX in real seawater samples, with a recovery ranging from 94.4% to 111%, good selectivity, stability, and repeatability. The authors suggested this kind of aptasensor to detect small molecules, but application and validation on real food samples should be highly recommended.

Nelis et al. proposed an enzyme-linked immunomagnetic electrochemical (ELIME) assay to detect domoic acid (DA) as a model target, utilizing screen-printed carbon electrodes (SPCEs), modifying with carbon black (CB) [53].

We remind that domoic acid (DA) is a marine toxin produced by phytoplankton species, *Nitzschia pungens*, and the main toxic agent associated with incidents of amnesic shellfish poisoning (ASP) on the east and west coasts of North America [54].

A comparison with SPCE pretreated by anodization (pre-SPCEs) and with SPCEs modified with other nanomaterials, such as gold nanospheres (GNS) and gold nanostars (GNST), was performed.

A competitive chronoamperometric immunosensor for the domoic acid (DA) was assembled using the differently modified SPCEs. Hapten-functionalized magnetic beads were used to avoid the individual SPCEs functionalization with antibodies. By comparison among the different modified electrodes, the CB-SPCE biosensor exhibited the best electroanalytical performances. DA was determined with a detection limit that is tenfold lower compared to pre-SPCE (4 vs. 0.4 ng mL^−1^). These results show very good agreement with HPLC data when analyzing contaminated scallops.

The method applied for detecting DA, using CB-SPCEs, showed great potential for the antibody-based determination of small molecules in a complex matrix.

Another known marine biotoxin produced by various dinoflagellates is okadaic acid. Chemically, OA is a polyether fatty acid derivative and exists in seafood, such as shellfish. The consumption of contaminated shellfish with OA leads to diarrheic shellfish poisoning (DSP), which results in the inhibition of protein phosphatase enzymes in humans.

Singh and coworkers have described the performances of an electrochemical microfluidic biochip to detect OA [55].

The screen-printed carbon electrode (SPCE) was modified by phosphorene–gold (BP–Au) nanocomposite, and an aptamer specific to OA was immobilized on it.

BP–Au nanocomposites were synthesized by an in situ, one-step method without using a reducing agent.

To improve the performances, a microfluidic platform was realized. The integrated system consisted of a microfluidic chip housing an aptamer modified SPCE as a single detection module for okadaic acid. The nanomaterials and the microfluidic channels prepared were spectroscopically and electrochemically analyzed. A detection limit of 8 pM and a linear range between 10 and 250 nM were obtained. Selectivity studies were also performed with mussel samples in the presence of interfering species. The aptasensor did not show any cross-reactivity with other types of food toxins.

Singh et al. developed a sensor based on a naphthalimide–gold nanocomposite to detect okadaic acid [56].

In this work, a composite for detecting OA using a naphthalimide-based receptor and gold nanoparticles were synthesized. The organic receptor was transformed into nanoparticles (ONPs) via the reprecipitation method. These ONPS were then coated on gold nanoparticles (Au@ONPs). The obtained composite was used to detect okadaic acid. UV-visible absorption spectroscopy, fluorescence spectroscopy and cyclic voltammetry techniques were used as the detection techniques, and a detection limit of 20 nM was obtained from UV-vis data.

The developed sensor maintained its sensing ability in the pH range of 5–9 and in high salt concentrations and was used for the OA detection in water samples.

As the most recent example of the detection of OA, we introduce an aptasensor developed by Lin [57].

A magnetic graphene-oxide (M-GO)-assisted homogeneous electrochemiluminescence (ECL) aptasensor was developed for sensitive detection of okadaic acid (OA). The aptamer and Ru(bpy)_3_
^2+^ were adsorbed in M-GO to prepare the ECL probe. The principle of M-GO-assisted homogeneous ECL aptasensor is illustrated in Figure 4.

When OA disassociated aptamer from M-GO, Ru(bpy)_3_
^2+^ was proportionally released from M-GO to generate the ECL signal. With the cooperation of deoxyribonuclease I (DNase I), the cyclic dissociation and degradation of aptamers induced much more available Ru(bpy)_3_
^2+^ for signal amplification. On the other hand, the unreleased Ru(bpy)_3_
^2+^ were still adsorbed in M-GO and magnetically separated. Hence, the background signal decreased, and the sensitivity was further improved. Results showed that the ECL intensity enhanced with the increasing logarithmic concentration of OA in the range of 0.01–10.0 ng mL^−1^, and the limit of detection was 4 pg mL^−1^.

The aptasensor has been used for OA detection in a real sample of mussels and represents a cost-effective approach for sensitive detection of marine toxins.

#### 3.1.2. Mycotoxins

The most common and abundant toxins present in nature are mycotoxins. They are produced by fungi [58] and can contaminate crops and foods, inducing teratogenic, mutagenic, carcinogenic, immunosuppressive, and endocrine-disrupting effects on humans and animals. To ensure food safety and prevent contamination risks in the agro-food sector, authorized levels for the most common mycotoxins in foods were established by the European Commission [59]. Therefore, various electrochemical biosensors using different analyzing techniques have been developed for mycotoxins monitoring at the required concentrations.

Zhang et al. [60] reviewed the newly released mycotoxin aptasensors, intending to provide indications concerning practical applications and tailored design of aptasensors for mycotoxins and other analytes.

More recently, Kong [61] reported the recent advances of different new immunosensors for mycotoxin determination over the past five years. The real application possibility, the advantages, and drawbacks, together with current challenges and future perspectives of these mycotoxin immunosensors, are evidenced.

Among the 400 different mycotoxins identified, aflatoxins presented high toxicity and carcinogenicity, and they are responsible for around 25% of animal mortality [58].

You et al. [62] reviewed the recent advances in electrochemical biosensors for aflatoxins detection, emphasizing the innovative sensing strategies based on electrochemistry, photoelectrochemistry, and electrochemiluminescence.

In the present review, some interesting examples of novel approaches and strategies to determine aflatoxins are reported and discussed.

Aflatoxins are detected in corn, peanuts, cottonseeds, nuts, almonds, figs, pistachios, spices, milk, and cheese and in various other food and beverages; they are stable at high temperatures. Consequently, they may resist the cooking processes [58]. Four types of aflatoxins were identified: AFB1, AFB2, AFG1, AFG2, plus two additional metabolites: AFM1 and AFM2, being AFB1 classified as the most abundant and toxic.

Among these, aflatoxin B1 (AFB1) is highly toxic, carcinogenic, mutagenic, genotoxic, and immunosuppressive, and classified as a group 1 carcinogen by International Agency for Research on Cancer (IARC) and a dose of more than 20 mg/kg bw (bw body weight) per day was associated with acute aflatoxicosis in adults [63].

An innovative electrochemical sensing strategy [64] was developed to detect AFB1 using aptamer (Apt)-complementary strands of aptamer (CSs) complex and exonuclease I (Exo I). A π-shape structure is organized on the surface of the electrode. The presence of π-shape structure as a double-layer physical barrier allowed the detection of AFB1 with high sensitivity. In the absence of AFB1, the π-shape structure remained intact, so only a weak peak current was recorded. Upon the addition of AFB1, the π-shape structure collapsed, and a strong current was recorded following the addition of Exo I. Under optimal conditions, a linear range between 7 and 500 pg mL^−1^ and a limit of detection of 2 pg mL^−1^ were observed. The developed aptasensor was also used to analyze AFB1 in spiked human serum and grape juice samples, and the recoveries were 95.4–108.1%.

Another strategy based on a competitive immunoassay using a secondary antibody conjugated with alkaline phosphatase enzyme as a tag was applied for the voltammetric detection of mycotoxins, ochratoxin (OTA) and AFM1, metabolite of AFB1, using modified gold screen-printed electrodes (AuSPEs) [65]. The biosensor was validated in red wine and milk samples with no need for pretreatment or preconcentration of the sample. The analytical signal was proportional to the toxin concentration in a wide linear range, showing a good limit of detection at the ng mL^−1^ level.

A magnetically assembled aptasensor [66] has been designed for label-free determination of AFB1 by employing a disposable screen-printed carbon electrode (SPCE) covered with a polydimethylsiloxane (PDMS) film as a micro electrolytic cell. The resulting label-free aptasensor has been developed using electrochemical impedance spectroscopy as an electroanalytical technique after the biorecognition between aptamers and the targets. The aptasensor showed a linear range from 20 pg mL^−1^ to 50 ng mL^−1^ with a detection limit of 15 pg mL^−1^ and was applied to detect AFB1 in spiked samples of peanuts. This sensing strategy seems to be a promising approach also for determining other targets.

An interesting AFB1 biosensor [67] is assembled by using a porous anodized alumina membrane modified with graphene oxide and an aptamer of AFB1. Briefly, the aptamer is immobilized on the surface of the porous anodized alumina nanochannels by covalent bonding. Graphene oxide is then immobilized on the surface by π–π stacking with the aptamer. On the addition of AFB1, graphene oxide is detached from the alumina surface because of the specific binding between AFB1 and the aptamer, resulting in the increased current response. The increase in current is proportional to the concentration of AB1. The detection limit of the aptasensor is about 0.13 ngmL^−1,^ and the linear range is from 1 to 20 ng mL^−1^. Furthermore, a good selectivity towards AFB1 was observed, but applying real food samples should be important for an effective sensor validation.

An electrochemical enzyme-linked oligonucleotide sensor for rapid detection of aflatoxin B1 (AFB1) is developed by Marrazza and coworkers [68].

The assay is based on a competitive format and disposable screen-printed cells (SPCs). Aflatoxin B1 conjugated with bovine serum albumin (AFB1-BSA) was immobilized by covalent binding on electropolymerized poly-(aniline–anthranilic acid) copolymer (PANI–PAA). After performing the affinity reaction between AFB1 and the biotinylated DNA-aptamer, the solution was dropped on the modified SPCs, and the competition occurred. The biotinylated complexes formed onto the sensor surface were coupled with a streptavidin–alkaline phosphatase conjugate. 1-naphthyl phosphate was used as an enzymatic substrate, and the electroactive product was detected by differential pulse voltammetry (DPV). A dose–response curve was obtained between 0.1 ng mL^−1^ and 10 ng mL^−1,^ and a limit of detection of 0.086 ng mL^−1^ was achieved. Finally, the sensor was applied for detecting AFB1 in maize flour samples.

Another electrochemical aptasensor achieving rapid detection of aflatoxin B1 (AFB1) was designed and developed by Zhao [69]. A short anti-AFB1 aptamer with a methylene blue (MB) as redox tag was immobilized on the surface of a gold electrode. Under optimized conditions, an AFB1 dynamic concentration range from 2 nM to 4 μM was obtained. The sensor could be well regenerated and reused. This sensor could detect AFB1 spiked in 20-fold diluted beer and 50-fold diluted white wine, respectively.

An electrochemiluminescence (ECL) platform based on a screen-printed bipolar electrode (BPE) was developed by Chen et al. [70] for sensitive detection of aflatoxin B1 in cereals.

The sensor included a cathode of closed BPE as a sensing interface and an anode as a signal collection interface. The BPE-ECL combination approach avoids the direct contact between the reaction/sensing system and the signal measurement system. In other words, the sensing system is physically separated from the signal measurements system. The authors argued that in this way, it is possible to eliminate the problem of false-positive and false-negative.

After mixing the test sample with a known concentration of horseradish peroxidase-labeled AFB1 (HRP-AFB1), a competition for binding to monoclonal antibodies occurred. The sensor showed a good analytical performance for AFB1 with a linear range from 0.1 to 100 ng mL^−1^ and a detection limit of 0.033 ng mL^−1^. Different kinds of cereals (rice, wheat, corn, sorghum, barley, and buckwheat) were selected as model grains to be tested. The results demonstrated that the recovery rate and accuracy of this sensor are at least comparable with those from ELISA.

A peculiar and innovative biosensor for the toxicity assessment of AFB1 and zearalenone (ZEN), another mycotoxin, was fabricated by Ghao et al. [71].

The International Agency for Research on Cancer (IARC) has classified ZEN as a group 3 substance (not carcinogenic to humans) [63], and the EFSA Panel on Contaminants in the Food Chain stated a tolerable daily intake (TDI) for ZEN of 0.25 mg/kg bw [72].

The proposed biosensor combines the advantages of both the electrochemical method and the peculiar characteristics of bacteria (*E. coli*) as the biorecognition element. The toxicity of mycotoxin AFB1 and ZEN are evaluated by the inhibition of *E. coli* metabolic activity. The combined toxic effect of the two mycotoxins was investigated, and synergistic biotoxicity was observed.

Under optimized experimental conditions, a linear concentration range of AFB1 and ZEN in the range of 0.01–0.3 and 0.05–0.5 mg mL^−1^, with the detection limits of 1 and 6 ng mL^−1^, respectively.

The recovery experiments in real oil samples (peanut and corn oils) indicated that the biosensor is applicable for the real sample mycotoxin detection.

An interesting strategy for AFB1 detection in grains [73] was based on DNA nanotetrahedron-structured probe (DTP), and horseradish peroxidase (HRP) triggered polyaniline (PANI) deposition. Briefly, the DNA nanotetrahedron was assembled on a gold electrode. Its carboxylic group was conjugated with the AFB1 monoclonal antibody (mAb) to form DTP. The test sample and a known set concentration of HRP-labeled AFB1 were mixed, and they compete for binding to DTP. The HRP assembled on the gold electrode catalyzed the polymerization of aniline on DTP. AFB1 in grains could be determined by using PANI, which could be detected using the electrochemical method. The dynamic AFB1 concentration range was from 0.05 to 20 ng mL^−1^. The detection limit was 0.033 ng mL^−1^. Rice, wheat, corn, sorghum, barley, and buckwheat were selected as model grains to be tested. The results showed that the recovery rate and accuracy of this sensor are comparable with those of ELISA. In fact, considering compared recovery data coming from the proposed method and the ELISA method, it can be deduced that the relative standard deviation ranged from −9.3% to 9.8%, which indicated there is no clear difference between the two data set.

Layer-by-layer self-assembly technology was used to assemble an electrochemical EIS aptasensor to detect AFB1 [74]. A multilayered sandwich structured electrode was obtained, depositing alternately positively charge layers (modified graphene nanosheets) and negatively charge layers (carboxylated polystyrene nanospheres). In this way, many electrochemical active sites and high conductivity were produced. The aptamer of AFB1 was immobilized on the positively charged layer via an amide bond. The optimized electrochemical aptasensor showed a limit of detection of 0.002 ng mL^−1^ and good stability after 30 days. The electrochemical aptasensor was applied to detect AFB1 in oil and soy sauce, yielding recovery values in the range of 94.5 and 103.3%.

A glassy carbon electrode (GCE) modified with a nanocomposite composed of poly-(4-aminobenzoic acid) (PABA), graphene oxide (GO), and gold nanoparticles (AuNps) was used for detecting AFB1 [75]. The carboxyl groups are used to bind covalently AFB1 antibodies via self-assembly of the antibody on AuNPs surface, enhancing the binding sites for the capture probe molecule and electrochemical signal. The obtained immunosensor showed a good linear range from 0.01 to 1 ng mL^−1^ and from 1 to 25 ng mL^−1^, and its detection limit is determined to be 0.001 ng mL^−1^. This immunosensor also demonstrated satisfactory reproducibility, selectivity, and stability. Moreover, the immunosensor could detect AFB1 in vegetable oil samples.

An electrochemical sensor based on a modified gold electrode to detect aflatoxin B1 (AFB1) [76] was assembled by using a 26-mer DNA aptamer with methylene blue (MB) label on an internal thymine (T) site (e.g., 18th T) and a thiol moiety at 5′ terminal. This sensor showed a detection limit of 6 pM and enabled detection of AFB1 in wine, milk, and corn flour samples. This sensor can be regenerated and shows good stability.

*Fusarium* mycotoxins are a general term for indicating the secondary metabolites produced by *Fusarium* species, and fumonisins is one the most representative family of this kind of mycotoxins.

Approximately 15 different derivatives of fumonisins have been discovered, including fumonisin A1 (FA1), FA2, FB1, FB2, FB3, FB4, FC1, FC2, FC3, FC4 and FP1 [77].

Fumonisin B1 (FB1) is the most toxic compound in this family, exhibiting hepato-, nephro-, and immunotoxicity in many animal species. It is also classified as group 2B carcinogen (possibly carcinogenic to humans) by the International Agency for Research on Cancer [63], and the EFSA Panel on Contaminants in the Food Chain stated a tolerable daily intake (TDI) for FB1 of 1.7 mg/kg bw [78].

Guo [79] reviewed the advances in biosensors, chemosensors, and assays based on the classical and novel recognition elements, such as antibodies, aptamers, and molecularly imprinted polymers. Application to food analysis, limits and time of the detection were also analyzed and discussed.

Some interesting examples of novel approaches and strategies to determine FB1 are reported and discussed in the following. We would like to underline that few examples of sensors to determine FB1 include electrochemical biosensors, probably because they are limited to dedicated applications because of the instability of their bioreceptors and fabrication difficulties. In this regard, there is still much room for improving FB1 determination.

Escarpa and his group [80] developed an electrochemical magneto immunosensor involving magnetic beads and disposable screen-printed carbon electrodes (SPCE) for fumonisins (FB1, FB2 and FB3). Once the immunochemical reactions took place on the magnetic beads, they were confined on the surface of SPCE, where electrochemical detection is achieved through the addition of suitable substrate and mediator for enzymatic tracer (Horseradish peroxidase, HRP). A detection limit of 0.33 μg L^−1^, good repeatability, reproducibility, and accuracy with a recovery rate of 85–96% was obtained. The magneto immunosensor was applied to fumonisin in beer samples with a good recovery rate of 87–105%.

Gunasekaran et al. [81] report an electrochemical immunosensing method for rapid and sensitive detection of two mycotoxins, fumonisin B1 (FB1) and deoxynivalenol (DON). A disposable screen-printed carbon electrode (SPCE) was used as a sensing platform. The working electrode was modified by gold nanoparticles (AuNPs) and polypyrrole (PPy)-electrochemically reduced graphene-oxide (ErGO) nanocomposite film. It can be considered a suitable platform for an effective anti-toxin antibody immobilization, with enhanced conductivity and biocompatibility.

Under optimized conditions, the limit of detection and linear range achieved for FB1 was 4.2 ppb and 0.2 to 4.5 ppm; and the corresponding values for DON were 8.6 ppb and 0.05 to 1 ppm. The immunosensor can specifically detect the two target toxins, even if present in the same sample. The sensor exhibited high sensitivity and low matrix interference when tested on spiked corn samples. Hence, this electrochemical immunosensing approach can be employed for the rapid detection of different mycotoxins present at the same in food.

As a more recent example, we would like to introduce a sensitive and selective electrochemical sensor using molecularly imprinted polymer nanoparticles (nanoMIPs) for FB1 recognition [82]. It is an electrochemical sensor, not properly a biosensor, but the detection strategy is very interesting and effective.

NanoMIPs were prepared by free-radical polymerization using the solid-phase synthesis method. The sensor was assembled in two steps. First, a film of the conducting polypyrrole–zinc porphyrin composite was deposited on a Pt electrode by electropolymerization. Then, nanoMIPs were covalently attached to this film. Both electrochemical impedance spectroscopy (EIS) and differential pulse voltammetry (DPV) were used for the sensor analytical characterization. The linear concentration range for FB1 was from 1 fM to 10 pM. The limit of detection was 0.03 and 0.7 fM, respectively. This electrochemical sensor showed no cross-reactivity vs. other mycotoxins. The FB1 recovery considering the FB1 spiked maize analysis samples was between 96 and 102%.

The last mycotoxins family we considered is that of ochratoxins, secondary metabolites secreted by fungi species (e.g., *Aspergillus* and *Penicillium*) during their growth. They are present in different crops and beverages, including coffee, wine, grape juice, and dried fruits [78]. Among them, ochratoxin A (OTA) is classified as a possible carcinogen by the International Agency for Research on Cancer (IARC) due to its severe toxicity [60], and the EFSA Panel on Contaminants in the Food Chain stated a tolerable daily intake (TDI) for OTA of 0.4 mg/kg bw [83]. In addition, OTA is chemically stable, so that it is metabolized very slowly with a half-life of more than 30 days in the body. With the recognition of its serious threat, developing smart sensing platforms for OTA plays a crucial role in food safety.

In a recent review [84], Wang reported an overview of the conventional and novel methods of OTA detection. The latest research progress and related applications of novel OTA electrochemical biosensors are mainly described with a new perspective. Furthermore, a summary of the current limitations and future challenges in OTA analysis is included, providing references for further research and applications.

Nevertheless, we reported and discussed some recent and interesting examples of OTA electrochemical detection.

As a first example, a label-free electrochemical impedimetric aptasensor for rapid detection and quantitation of OTA in cocoa beans is reported [85]. The anti-OTA aptamer was immobilized on screen-printed carbon electrodes (SPCEs) via a diazonium-coupling reaction. The aptasensor exhibited a limit of detection of 0.15 ng/mL, showed good selectivity and reproducibility. The increase in electron transfer resistance was linearly proportional to the OTA concentration in the range 0.15–2.5 ng mL^−1^, with a recovery percentage of 91–95%, obtained in cocoa samples. The analysis can be performed on-site employing a portable EIS setup.

Another impedimetric aptasensor able to directly detect OTA without any amplification procedure has always been developed by Marty and his group [86]. This aptasensor was assembled by coating the surface of a gold electrode with a film of polypyrrole (PPy), modified with covalently bound polyamidoamine dendrimers of the fourth generation (PAMAM G4). Finally, DNA aptamers binding, specifically OTA, were covalently bound to the PAMAM G4. The OTA detection was performed using electrochemical impedance spectroscopy (EIS), and the results indicated that the presence of OTA led to the modification of the electrical properties of the PPy film due to the aptamers’ conformational changes after the OTA-specific binding. The aptasensor had a dynamic range of up to 5 mg L^−1^ of OTA and a detection limit of 2 ng L^−1^ of OTA, which is below the OTA concentration authorized in food by the European legislation. The efficient detection of OTA by this electrochemical aptasensor provides a platform that can be used to detect various small molecules through specific aptamer associations.

Marty group proposed another sensor for ochratoxin A (OTA) detection in cocoa beans using a competitive aptasensor and differential pulse voltammetry (DPV) [87]. In this case, biotin-labeled and free OTA competed to bind with immobilized aptamer onto the surface of a screen-printed carbon electrode (SPCE). The developed aptasensor showed good linearity in the range 0.15–5 ng mL^−1^ with the limit of detection of 0.07 ng mL^−1^. The aptasensor displayed good recovery values in the range 82.1–85%, thus, showing its efficiency for complex matrices.

An impedimetric label-free immunosensor to always detect OTA in cocoa beans is reported by Albanese and coworkers [88].

In this paper, two different approaches of anti-OTA immobilization are involved, considering that the immobilization method on conductive surfaces plays a central role in the immunosensor performances [89]. The method for the immobilization in oriented mode consisted of orienting the interacting sites (Fab) of antibodies towards the test solution containing the antigen molecules. Fab is fragment antigen-binding, i.e., a region of an antibody that binds to the antigen.

The “orienting” agent is protein A/G, an immunoglobulin (Ig)-binding protein, showing specificity for the heavy chains of the Fc region of antibodies, thus effectively orienting the immobilized antibodies with antigen-binding sites outward-looking. If the protein A/G is not used, the anti-OTA immobilization occurs in a nonspecific position, and the resulting interaction between OTA and anti-OTA is less effective.

It is well-known Fc region is the fragment crystallizable region, i.e., the tail region of an antibody that interacts with cell surface receptors called Fc receptors and some proteins of the complement system. This property allows antibodies to activate the immune system.

The performances of the two antibody immobilization methods (oriented and not oriented) were compared, highlighting a lower limit of detection (5 pg mL^−1^) for the not oriented immobilization and a shorter linear range than that of the oriented immunosensors, which showed linearity range from 0.01 to 5 ng mL^−1^ OTA. Using atomic force microscopy (AFM) clarified that the oriented immobilization created a more ordered and highly dense antibody surface.

Finally, the oriented immunosensor was used to determine OTA in spiked cocoa beans samples, and the results were compared with those recorded with a competitive ELISA kit. The immunosensor was sensitive to OTA levels lower than 2 mg kg^−1^, representing the lower acceptable limit for OTA according to the European legislation for the common food products.

A sensitive electrochemical aptasensor for OTA was successfully assembled by Wang [90]. This aptasensor was prepared to combine a nanocomposite of gold nanoparticles (AuNPs) functionalized with silica-coated iron-oxide magnetic nanoparticles (mSiO2@Au) with another nanocomposite, including cadmium telluride quantum dots (CdTe QDs), graphene and AuNPs (GAu/CdTe). The aptasensor exhibited a linear range from 0.2 pg mL^−1^ to 4 ng mL^−1^ and a detection limit of 0.07 pg mL^−1^.

This work provides a novel strategy for sensitive detection of various target molecules and would have great potential in food safety monitoring and clinical diagnosis, but no analysis on real samples has been provided.

A label-free electrochemical OTA aptasensor was realized by Yang [91], taking advantage of the intrinsic peroxidase-like activity of graphite-like carbon nitride (g-C_3_N_4_) nanosheet (g-CNNS) and its high-affinity towards single-strand DNA.

This aptasensor did not require labeled aptamer and immobilization of g-CNNS compared with previous g-CNNS-based aptasensors. As a result, this aptasensor showed a detection limit of 0.073 nM and was employed to assay OTA in the real samples, including red wines, juices, and corns.

A sensitive signal-on electrochemical aptasensor has been proposed [92] for OTA detection, based on DNA-controlled layer-by-layer assembly of dual gold nanoparticle (AuNP) conjugates.

Both qualitative and quantitative analyses of OTA were thus realized by differential pulse voltammetry (DPV) signals, with a detection limit of 0.001 ppb and a dynamic range from 0.001 to 500 ppb over 6 orders of magnitude. Moreover, the real sample analysis towards OTA spiked wine samples showed good recovery results. This sensing platform can represent a promising system for food routine quality control.

A green electrochemical immunosensor to detect OTA was prepared [93] by self-assembling a 2-mercaptoacetic (TGA) monolayer on the surface of the working Au electrodes to assemble the Au/TGA/bovine serum albumin (BSA)-OTA/anti-OTA monoclonal antibody composite probe for selective and sensitive detection of OTA. The immunosensor detection approach is based on indirect competitive principle and differential pulse voltammetry analysis.

Under the optimal conditions, the developed immunosensor showed a limit of detection of 0.08 ng mL^−1^ in the range of 0.1 and 1.0 ng mL^−1^ for OTA.

Real application in the spiked malt samples showed high accuracy with no matrix interferences for the proposed immunosensor.

**Table 1 molecules-26-02940-t001:** An overview of recent electrochemical biosensors for toxins determination.

Electrode	(Bio)Sensor Format	Electrochemical Technique	Analyte/Sample	L.R.	LOD	References
SPCEs	Label-free electrochemical aptasensor based on DNA nanotetrahedron and DNA triplex	SWV	Saxitoxin/seawater	1–400 nM	0.92 nM	[43]
CB-SPCEs	Enzyme-linked immunomagnetic electrochemical (ELIME) assay	CA	DA/shellfish scallop	5–62 ng mL^−1^	0.4 ng mL^−1^	[53]
Phosphorene–gold–SPCE (BP–AuSPCE)	Electrochemical microfluidic biochip, including BP-SPCE with an OA aptamer	DPV	OA/mussels	10–250 nM	8 pM	[55]
Indium–tin-oxide electrode (ITO)	Electrochemiluminescence (ECL) aptasensor supported by magnetic graphene oxide (M-GO	ECL/CV	OA/mussels	0.01–10 ng mL^−1^	4 pg mL^−1^	[57]
AuSPE	Electrochemical aptasensor based on aptamer-complementary strands of aptamer complex forming a π-shape structure on the surface of the electrode and exonuclease I (Exo I)	DPV	AFB1/human serum, grape juice	7–500 pg mL^−1^	2 pg mL^−1^	[64]
AuSPE	Electrochemical immunosensor utilizing a competitive assay format	DPV	OTA and AFM1/red wine, milk	-	OTA 15 ng mL^−1^ AFM1 37 ng mL^−1^	[65]
SPCE	Magnetically assembled aptasensor for label-free determination of AFB1 employing a disposable screen-printed carbon electrode (SPCE) covered with polydimethylsiloxane (PDMS) film as the micro electrolytic cell	EIS	AFB1/peanuts	20–50 pg mL^−1^	15 pg mL^−1^	[66]
GO–PAA	Aptasensor employing PAA modified with GO and an aptamer of AFB1	Amperometry	AFB1/no real samples	1–20 ng mL^−1^	0.13 ng mL^−1^	[67]
SPCEs	Aptasensor using a competitive format and modified screen-printed electrode	DPV	AFB1/maize flour	Dose–response curve 0.1–10 ng mL^−1^	0.086 ng mL^−1^	[68]
AuE	Aptasensor having methylene blue (MB) as redox tag	SWV	AFB1/white wine	2 nM–4 μM	2 nM	[69]
Screen-printed bipolar electrode (BPE)	BPE-ECL aptasensor	ECL	AFB1/rice, wheat, corn, sorghum, barley, and buckwheat grains	0.1–100 ng mL^−1^	0.033 ng mL^−1^	[70]
GCE	Biosensor for AFB1 and ZEN using *Escherichia coli* as biorecognition element	CA	AFB1 and ZEN/peanut and corn oil	AFB1 0.01–0.3 mg mL^−1^ ZEN 0.05–0.5 mg mL^−1^	AFB1 1 ng mL^−1^ ZEN 6 ng mL^−1^	[71]
AuE	Immunosensor based on DNA tetrahedron-structured probe (DTP), obtained from the conjugation between DNA tetrahedron nanostructures and HRP -labeled AFB1 monoclonal antibody	DPV	AFB1/rice, wheat, corn, sorghum, barley, and buckwheat grains	0.05–20 ng mL^−1^.	0.033 ng mL^−1^	[73]
LbL-GCE	Aptasensor assembled via layer-by-layer deposition of differently charged layers onto GCE. The AFB1 aptamer was immobilized onto the negatively charged layer	EIS	AFB1/oil and soy sauce	0.001–0.10 ng mL^−1^	0.002 ng mL^−1^	[74]
AuNPs–GO–PABA–GCE	Immunosensor where AFB1 antibodies are linked to AuNPs–GO–PABA nanocomposite, deposited on GCE	EIS	AFB1/vegetable oils	0.01–1.0 ng mL^−1^; 1–25 ng mL^−1^	0.001 ng mL^−1^	[75]
AuE	Aptasensor where AFB1 aptamer is immobilized onto MCH layer self-assembled on AuE	SWV	AFB1/wine, milk, corn flour	8 pM–25 nM; 25 nM–3 μM	6 pM	[76]
MBs–SPCEs	Electrochemical magnetoimmunosensor involving magnetic beads (MBs) and disposable carbon screen-printed electrodes (SPCEs)	Amperometry	FB1/beer	Nonlinear calibration curves performed	0.33 mg L^−1^	[80]
AuNPs–PPy–rGO–SPCEs	Immunosensor using AuNPs–PPy–rGO nanocomposite as a platform for immobilizing anti-toxin antibody	DPV	FB1, DON/corn	FB1 0.2–4.5 ppm; DON 0.05–1 ppm	FB1 4.2 ppb; DON 8.6 ppb	[81]
NanoMIPs–PPY–ZnP–Pt	Chemosensor based on nano imprinted polymer nanoparticles (nano MIPs) immobilized	DPV, EIS	FB1/maize flour	1 fM–10 pM	EIS 0.7 fM; DPV 0.03 fM	[82]
SPCE	Label-free electrochemical impedimetric aptasensor based on the diazonium-coupling reaction mechanism for the immobilization of anti-OTA aptamer at SPCEs	EIS	OTA/cocoa beans	0.15–2.5 ng mL^−1^	0.15 ng mL^−1^	[85]
AuE	Aptasensor based on the modified gold electrode with conductive polypyrrole layer covalently bound to polyamidoamine dendrimers of the fourth generation (PAMAM G4), where the OTA aptamer was immobilized	EIS	OTA/wine	-	2 ng L^−1^	[86]
SPCE	Competitive aptasensor where biotin-labeled and free OTA compete to bind with immobilized aptamer onto the surface of a screen-printed carbon electrode (SPCE)	DPV	OTA/cocoa beans	0.15–5 ng mL^−1^	0.07 ng mL^−1^	[87]
Au thin-film single electrodes	Impedimetric label-free immunosensor using two antibody immobilization methods (oriented, including protein A/G and not oriented)	EIS	OTA/cocoa beans	Oriented 0.01–5 ng mL^−1^ Not oriented 5 × 10^−3^–0.05 ng mL^−1^	Oriented 0.01 ng mL^−1^ Not oriented 5 × 10^−3^ ng mL^−1^	[88]
Bismuth-coated glassy carbon electrode (BFE)	Aptasensor assembled by combining nanocomposites of gold nanoparticles (AuNPs) functionalized silica-coated iron-oxide magnetic nanoparticles (mSiO2@Au) and cadmium telluride quantum dots (CdTe QDs) modified graphene/AuNPs nanocomposites (AuNPs/CdTe)	SWV	OTA/no real samples	0.2–4 ng mL^−1^	0.07 pg mL^−1^	[90]
AuE	Label-free electrochemical OTA aptasensor based on the peroxidase-like activity of g-C3N4 nanosheet (g-CNNS) and its high affinity toward single-strand DNA	CV	OTA/red wines, juices, corns	0.2–500 nM	0.073 nM,	[91]
AuE	Signal-on electrochemical aptasensor for OTA assay based on DNA controlled layer-by-layer assembly of dual gold nanoparticle (AuNP) conjugates	DPV	OTA/wine	0.001–500 ppb	0.001 ppb	[92]
TGA–AuE	Electrochemical immunosensor based on self-assembling a 2-mercaptoacetic (TGA) monolayer on the surface of Au electrode to form the Au/TGA/bovine serum albumin (BSA)-OTA/anti-OTA monoclonal antibody composite probe	DPV	OTA/malt	0.1–1.0 ng mL^−1^	0.08 ng mL^−1^	[93]

Abbreviations: AFB1: aflatoxin B1; AuE: gold electrode; AuNPs: gold nanoparticles; AuSPE: gold screen-printed electrode; BFE: bismuth-coated glassy carbon electrode; BPE: bipolar electrode; CA: chronoamperometry; CB: carbon black; CV: cyclic voltammetry; DPV: differential pulse voltammetry; EIS: electrochemical impedance spectroscopy; ECL: electrochemiluminescence; FB1 fumonisin B1; rGO: reduced graphene oxide; GO: graphene oxide; ITO: indium–tin-oxide electrode; lbL: layer-by-layer; MBs: magnetic beads; MIPs: molecularly imprinted polymers: m-GEC: magnetic graphite-epoxy composite; MWCNTs: multi-walled carbon nanotubes; OA: okadaic acid; OTA: ochratoxin; PANI: polyaniline; PAA: poly(anthranilic acid); PPY: polypyrrole; QDs: quantum dots; SPCE: screen-printed carbon electrode; STX: Saxitoxin; SWV: square-wave voltammetry; SWASV: square-wave anodic stripping voltammetry; TGA: 2-mercaptoacetic acid.

### 3.2. Pathogenic Bacteria

Bacteria are the most common cause of foodborne diseases in the world [94]. Due to the potential threat of foodborne pathogens and the fact that the infective dose of some of them is low, pathogenic cells of some species must be totally absent from food. For example, see the *Salmonella* case [95].

Considering all these critical issues, developing accurate, simple, rapid, low-cost, and possibly portable devices able to make point-of-care analyses is mandatory. Biosensors seem to be suitable analytical tools, complying with most all these requirements.

Du [96] reviewed recent developments in electrochemical biosensing technologies used to detect common foodborne pathogens, evidencing that biosensing technology is a sufficiently mature technology to be applied to the determination of pathogenic bacteria.

Riu [94] also reviewed novel electrochemical biosensors for pathogenic bacteria, providing a critical overview about the state-of-art of biosensors and, at the same time, some trends and indications for future developments in this area.

The present review focuses on the most recent advances in electrochemical (bio)sensors to detect pathogenic bacteria in food. Papers published in the last 5–6 years are reviewed, and Table 2 summarizes the analytical characteristics of recent electrochemical biosensors for pathogenic bacteria reported in the review.

There are many kinds of pathogens producing toxins causing foodborne diseases [97], among them *Escherichia coli*, *Staphylococcus aureus*, *Salmonella*, and *Listeria monocytogenes* are common.

#### 3.2.1. Salmonella

*Salmonella* is a species of rod-shaped Gram-negative bacteria belonging to the family of Enterobacteriaceae. It contains two main species, *Salmonella enterica* and *Salmonella bongori,* with more than 2500 serotypes, and all these serotypes can cause disease in humans [98].

WHO declared *Salmonella* as one of the four major global causes of diarrheal diseases and one of the pathogenic bacteria with emergent resistant serotypes [99].

Considering all these criticalities, including extremely low infection limits (1 CFU), the levels of *Salmonella* in food-regulated by-laws have been tightened over the years. For example, European Commission [97] required the absence of *Salmonella* in a defined amount of a food product (e.g., 10 or 25 g) placed on the market during the shelf life.

Magalhães [98] has reviewed the commercially available rapid methods for *Salmonella* detection. The potentialities of electrochemical biosensors for developing rapid devices are highlighted. The state-of-art and the newest and innovative technologic approaches are presented, and a critical analysis of the literature has been carried out, evidencing the current challenges towards a complete solution of the *Salmonella* detection criticalities.

More recently, Li [99] has presented a more general overview on *Salmonella* biosensors by highlighting the different typologies (optical, electrochemical, piezoelectric, etc., biosensors) and analyzing recent trends, particularly the integration with nanomaterials, microfluidics, portable instruments, and smartphones.

Nevertheless, we reported and discussed some innovative and interesting examples of *Salmonella* electrochemical biosensors. Generally, the detection of *Salmonella enterica* serotype Typhimurium is considered, so, for this reason, and for reasons of brevity, it is referred to as *Salmonella*, unless otherwise stated.

Kraatz and coworkers [100] have developed an electrochemical immunosensor to detect *Salmonella* based on a glassy carbon electrode modified with high-density gold nanoparticles (AuNps) well dispersed in chitosan hydrogel. The composite film has been used as a platform for the immobilization of biorecognition elements, such as the capture anti-body (Ab1). A sandwich electrochemical immunosensor has been assembled after incubation with *Salmonella* and the horseradish peroxidase (HRP) *Salmonella* secondary anti-body (Ab2). The immunosensor showed a linear range from 10 to 10^5^ CFU mL^−1^ with a low detection limit of 5 CFU mL^−1^. Furthermore, the sensor’s performances in real-to-life conditions were tested by analyzing tap water and milk samples containing *Salmonella*. The results were compared and validated with those obtained by the plate counting method, indicating that the immunosensor is suitable for food safety analysis.

A label-free impedimetric aptamer-based biosensor [101] for *Salmonella* detection was prepared by grafting a diazonium-supporting layer onto screen-printed carbon electrodes (SPCEs) because this procedure allowed forming a denser aptamer layer, resulting in a higher sensitivity.

The developed aptamer-biosensor responded linearly, on a logarithm scale, over the concentration range from 1 × 10^1^ to 1 × 10^8^ CFU mL^−1^, with a limit of detection of 6 CFU mL^−1^. Selectivity studies showed that the aptamer biosensor could discriminate *Salmonella* from six other model bacteria. Finally, the aptamer biosensor was applied to the *Salmonella* detection in spiked apple juice samples with good recovery results.

Another label-free impedimetric biosensor [102] to detect *Salmonella* was developed based, this time, on combining poly-[pyrrole-co-3-carboxyl-pyrrole] copolymer and aptamer. The aptamer/target interaction on the conjugated copolymer and the copolymer conductivity modification improved the impedimetric measurements.

In fact, when the bacteria is present and interacts with the aptamer, a clear increase of the EIS response is observed. This interaction between the aptamer and the bacteria produced significant changes in the aptamer–copolymer film conductivity and in the electrical properties by modifying the environment considerably near the polymer chains, i.e., by modifying p–p conjugation, also produced changes in the interfacial double layer due to the different charge properties of the pathogens and decrease of the polymeric dopant mobility due to the electrostatic repulsion of the pathogens. The aptasensor detected *Salmonella* in the concentration range of 1 × 10^2^–1 × 10^8^ CFU mL^−1^ with good selectivity vs. other model pathogens and with a limit of detection of 3 CFU mL^−1^. Finally, like the previous example [101], the aptamer biosensor was applied to the *Salmonella* detection in spiked apple juice samples with good recovery results.

An electrochemical aptasensor [103] was assembled employing a thiol functionalized aptamer-immobilized onto the electrochemically reduced graphene-oxide–chitosan composite (rGO–CHI) as a conductive platform for the *Salmonella* detection.

The sensitivity and selectivity of this aptasensor against the pathogen target were evaluated using cyclic voltammetry and differential pulse voltammetry. The developed aptasensor is specific to *Salmonella* and can distinguish *Salmonella* from other pathogens. The aptasensor showed a low limit of detection of 1 × 10^1^ CFU mL^−1^. The sensor was applied to artificially contaminated raw chicken samples, and the results were coherent with those obtained from pure cultures.

A highly sensitive and specific electrochemical aptasensor [104] for *Salmonella* detection was developed by combining target-induced aptamer displacement on gold nanoparticle (AuNP)-modified gold electrodes with rolling circle amplification (RCA). The sensor showed a detection limit of 16 CFU mL^−1^ and a linear detection range from 20 to 2 × 10^8^ CFU mL^−1^ and also demonstrated acceptable reproducibility and low matrix effect.

The proposed strategy was further applied to some real samples for evaluating the recovery, so different concentrations of *Salmonella* were spiked into bottled mineral water and into pure milk. The recovery results are good for both sample typologies.

A *Salmonella* biosensor able to process a large sample volume [105] was developed by Capobianco by using an Ag/AgCl reference electrode, a platinum counter electrode, and a porous working electrode made from graphite felt coated with antibodies specific for *Salmonella* antigens. This design allows samples to flow through the electrode while capturing target pathogens.

The detection limit of 1000 *Salmonella* cells was obtained in samples with a volume of 60 mL. The low-cost sensor allows for incorporation into disposable detection devices, but an evaluation of the sensor analytical parameters on real samples, recovery included, should be very useful.

Korecka et al. [106] developed a fast and efficient biosensor for the screening of milk samples contaminated by *Salmonella*. A smart approach was performed where bacterial cells were separated immunomagnetically, with subsequent reaction with conjugate, i.e., specific IgG molecule labeled with an electrochemical indicator.

The peculiar structure of this indicator included hyperbranched dendron polymeric molecules and heavy metal quantum dots (QDs). Square-wave anodic stripping voltammetry (SWASV) using screen-printed carbon electrodes modified with in situ formed Bi (III) film (BiSPCE) was used for determining the metal ions released from the QDs (CdTe). The metal ion signals can be correlated to the number of detected bacteria cells. By this method, the *Salmonella* samples were analyzed in 2.5 h, even evidencing even a minimal number of bacterial cells (4 CFU) in 1 mL of the sample. The whole system was verified using real food samples. UHT whole milk samples were artificially spiked, and the obtained results are very promising. In fact, this methodological approach seems to perform analyses in a limited sample volume, detecting few CFU/mL. Moreover, this sensor can be considered as an interesting tool for rapid bacteria screening in milk and changing the used antibodies. It seems a flexible tool for detecting other common pathogens and a possible candidate for a multi-pathogen detection.

A sensitive electrochemical aptasensor [107] was developed by Lin and coworkers using aptamer-coated gold-interdigitated microelectrode for target capture, and impedance measurement and antibody-modified nickel nanowires (NiNWs) for target separation and impedance amplification.

A linear concentration range was obtained from 1 × 10^2^ to 1 × 10^6^ CFU mL^−1^ in 2 h with the detection limit of 80 CFU mL^−1^. The mean recovery for the spiked chicken samples was 103.2%, and it can be considered acceptable.

In a more recent paper, Lin [108] designed and assembled another impedance biosensor always to detect *Salmonella* using multiple magnetic nanobeads (MNB) nets in a ring channel for continuous-flow separation of bacteria cell from 10 mL of sample, manganese dioxide nanoflowers (MnO_2_ NFs) as a nanomaterial for biological signal amplification, and an interdigitated microelectrode for sensitive impedance measurements. The approach is comparable with the other illustrated in the previous paper [107], but the role of MNB nets is new since they act as separate elements of the target bacteria.

This biosensor could separate ~60% of *Salmonella* from 10 mL of bacterial sample and detect *Salmonella* with a linear range of 3.0 × 10^1^ to 3.0 × 10^6^ CFU mL^−1^ in 1.5 h with a lower detection limit of 19 CFU mL^−1^. Moreover, this biosensor was evaluated by detecting the target bacteria in spiked chicken meat samples, and the results are comparable with those obtained on the same samples with the plate counting method, indicating that it is suitable for food safety analysis.

The last impedance biosensor for *Salmonella* developed by the Lin group [109] is based on rotary magnetic separation and cascade reaction. First, magnetic nanoparticles (MNPs) modified with anti-*Salmonella* monoclonal antibodies were injected into a capillary in the presence of a rotary gradient magnetic field. Then, a bacterial sample was injected into the capillary, and the target bacteria were continuous-flow captured onto the MNPs. When organic–inorganic hybrid nanoflowers were prepared using manganese dioxide (MnO_2_), glucose oxidase (GOx) and anti-*Salmonella* polyclonal antibodies (pAbs), they were injected to label the bacteria, resulting in forming MNP–bacteria–nanoflower sandwich complexes. Finally, glucose (low conductivity) was injected and oxidized by GOx on the complexes to produce H_2_O_2_ (low conductivity) and gluconic acid (high conductivity), leading to an impedance decrease. Moreover, the produced H_2_O_2_ triggered a cascade reduction of MnO_2_ into Mn^2+^, leading to further impedance decrease. The impedance changes were measured using an interdigitated microelectrode and correlated to the concentration of target bacteria. This biosensor could detect *Salmonella* ranging from 1 × 10^1^ to 1 × 10^6^ CFU mL^−1^ in 2 h with a low detection limit of 10^1^ CFU mL^−1^ and a mean recovery of 100.1% for the spiked chicken samples. Considering the last three papers, the basic approach is very similar. On the other hand, a real comparison, not only evaluating analytical performances but also costs, specificity and reproducibility, is mandatory.

#### 3.2.2. Escherichia coli

In addition to *Salmonella,* another pathogenic bacterium commonly associated with foodborne outbreaks is *Escherichia coli*. The infection is usually acquired via the fecal–oral route by consuming contaminated and raw food, such as beef, various leaf vegetables, unpasteurized milk, and water. It should be underlined that food/waterborne diseases due to *E. coli* are among the major causes of illness in many developing countries [110], causing gastroenteritis and related diseases leading to dramatic consequences [111].

Recently, in 2017, Li and coworkers have reviewed [112] the advancements in developing electrochemical biosensors for the rapid detection of *Escherichia coli*, illustrating the different configurations of biosensors and the sensing approaches.

This review reports some interesting and innovative examples of *E. coli* electrochemical biosensors developed in the last 5–6 years.

The electrochemical genosensor is one of the most promising methods for the rapid and reliable detection of pathogenic bacteria, and an electrochemical genosensor was developed by Sun for *E. coli* detection [113].

The genosensor included a gold electrode where complementary DNA was immobilized on a SAM, hybridizing with a specific fragment pathogen gene to build a sandwich structure. Multiwalled carbon nanotubes (MWCNT), embedded in chitosan with a layer of bismuth, modified a GCE for detecting the performance of the sensor.

The detection limit was 1.97 × 10^−14^ M. The genosensor showed good sensitivity and selectivity, and it was also applied for determining the pathogen in real beef samples contaminated artificially.

A magneto-genosensing approach for detecting the three most common pathogenic bacteria in food safety, such as *Salmonella*, *Listeria* and *Escherichia coli*, is developed by the Alegret group [114]. The methodology is based on detecting the tagged amplified DNA obtained by single-tagging PCR with specific primers for each pathogen, followed by electrochemical magneto-genosensing on silica magnetic particles (silica MPs). A set of primers was selected to amplify the primers for each set tagged with fluorescein, biotin and digoxigenin coding for *Salmonella enterica*, *Listeria monocytogenes* and *E. coli*, respectively. The single-tagged amplicons were then immobilized on silica magnetic particles based on the nucleic acid-binding properties of silica particles in the presence of the chaotropic agent, such as guanidinium thiocyanate. The assessment of the silica MPs as a platform for electrochemical magneto-genosensing is described. A linear concentration range from 0.03 to 3 ng mL^−1^ was observed with the detection limits of 0.04, 0.13 and 0.05 ng mL^−1^ for *S. enterica*, *L. monocytogenes* and *E. coli*, respectively. It should be noticed that an evaluation of the sensor analytical parameters on real samples, recovery included, resulted in very useful, at least for one of the three pathogens.

Graphene wrapped copper (II)-assisted cysteine hierarchical structure (rGO–CysCu) has been used as a sensing layer to assemble an impedimetric label-free electrochemical immunosensor for the quantitative determination of *Escherichia coli* [115].

Under optimal conditions, the calibration plots were linear in the detection range from 10 CFU mL^−1^ to 1 × 10^8^ CFU mL^−1^ with a detection limit of 3.8 CFU mL^−1^. Moreover, the proposed immunosensor showed good selectivity and specificity towards the nonpathogenic *E. coli* and other bacterial cells in the synthetic samples. The validation of the immunosensor was carried out using artificially contaminated real samples (*E. coli* spiked tap water, juices, and skimmed milk), and the results are comparable with those obtained using the plate count method.

Wang has assembled [116] all-solid-state luminol-electrochemiluminescence (ECL) *Escherichia coli* aptasensors by using AgBr nanoparticles/3D nitrogen-doped graphene hydrogel (AgBr/3DNGH).

The multifunctional nanoarchitecture was used as an all-solid-state ECL platform for assembling an *E. coli* aptasensor via glutaraldehyde as a crosslinker between amine-functionalized *E. coli* aptamer and luminol/AgBr/3DNGH. Since *E. coli* can significantly decrease the ECL intensity because of the steric hindrance mechanism, the proposed aptasensor displayed a linear response for *E. coli* in the range from 0.5 to 500 CFU mL^−1^ with a detection limit of 0.17 CFU mL^−1^. In Figure 5, the preparation steps and detailed measurement sequence of the aptasensor is illustrated.

To further demonstrate the applicability of the proposed aptasensor, the recovery test was performed by the standard addition method. Different concentrations of *E. coli* were added into meal samples. The recovery rate ranged from 99.4% to 101.2%, indicating that the proposed method was stable and could be applied to analyze real samples.

A bridged rebar graphene (BRG) functionalized label-free impedimetric aptasensor for *E. coli* detection was developed by Sabherwal et al. [117]. BRG was synthesized by chemical unscrolling MWCNT for producing graphene, followed by bridging with terephthalaldehyde (TPA) to form a 3D hierarchical nanostructure. A scheme related to the aptasensor assembling and sensing approach is illustrated in Figure 6.

The developed nanostructured aptasensor demonstrated a low detection limit of 10 CFU mL^−1^ with a dynamic response range from 10 to 1 × 10^6^ CFU mL^−1^ in spiked water, juice, and milk samples.

An immunosensor based on a hybrid nanocomposite composed of poly(pyrrole), gold nanoparticles, multiwalled carbon nanotubes and chitosan (PPy/AuNP/MWCNT/Ch) was developed [118]. This hybrid nanocomposite modified a pencil graphite electrode (PGE) and the anti-*E. coli* monoclonal antibody was immobilized on the resulting platform.

Under the optimum conditions, concentrations of *E. coli* from 3 × 10^1^ to 3 × 10^7^ CFU/mL^−1^ were detected with a detection limit of 30 CFU mL^−1^ in PBS buffer. Good results in terms of specificity and stability were also achieved.

On the other hand, it should be mentioned that evaluating the sensor analytical parameters on real samples was very useful.

Dou [119] has reported the assembling of a nonenzymatic sandwich-type electrochemical immunoassay for quantitative monitoring of *Escherichia coli*. Silica-coated Fe_3_O_4_ magnetic nanoparticles (Fe_3_O_4_@SiO_2_) were modified with mouse anti-*E. coli* monoclonal antibody (Ab1) as the capture probes reducing the measurement time and increasing the sensitivity. Au@Pt nanoparticles were loaded on neutral red (NR) functionalized graphene producing a nanocomposite rGO–NR–Au@Pt with high specific surface area and good biocompatibility and acting as a carrier of the detection antibodies (Ab2).

Under the optimized conditions, a linear concentration range is from 4.0 × 10^3^ to 4.0 × 10^8^ CFU mL^−1,^ and the limit of detection is 4.5 × 10^2^ CFU mL^−1^. The immunoassay showed acceptable specificity, reproducibility and good performance in terms of recovery, analyzing spiked commercial pork and milk samples.

Recently, Capobianco [120] proposed a flowthrough immunoelectrochemical biosensor for *E. coli* detection, taking advantage of the same sensing approach for determining *Salmonella* [105]. As previously reported [105], the working electrode was a porous, antibody-coated graphite felt electrode acting as both a biorecognition element coated for capturing target pathogen as well as a signal transducer.

The low detection limit for a sample containing 10,000 *E. coli* cells in 5, 60, and 1000 mL of buffer was 2000, 170, and 10 cells mL^−1^, respectively, in a total assay time of 3 h, whereas the low detection limit for *E. coli* was 400 cells mL^−1^ in spiked beef samples.

#### 3.2.3. Staphylococcus aureus

*Staphylococcus aureus* is one of the most common foodborne pathogens, and its infections can cause even more deaths than AIDS, tuberculosis and viral hepatitis combined [121].

Oh [121] reviewed the state-of-the-art of biosensing approaches and methodologies for detecting *S. aureus*, illustrating the most used ones based on different transducing modes, such as electrochemical, optical, and mass-based biosensors.

Herein, we focused on the most recent developments of the electrochemical biosensors for *S. aureus* determination, providing some interesting examples.

An electrochemical biosensor for rapid detection of *S. aureus* based on the silver wire across electrodes was reported by He [122].

Fragment of bacterial 16S rRNA hyper-variable region was used as a biomarker and hybridization chain reaction (HCR) combined with silver deposition technique was used to form silver wire crossing electrode.

A multichannel series piezoelectric quartz crystal (MSPQC) was utilized as a detector. Using the reported 16S rRNA fragment for the *Staphylococcus aureus* determination, a linear concentration range from 50 to 10^7^ CFU mL^−1^ within 100 min was obtained. The detection limit was 50 CFU mL^−1^.

Moreover, the proposed biosensor showed good selectivity and specificity towards other bacteria and pathogens, such as *Escherichia coli*, *Salmonella* enteritidis, *Listeria* innocua, *Pseudomonas aeruginosa* and Streptococcus pneumoniae. Artificially contaminated human serum samples and milk samples were analyzed with the proposed biosensor, obtaining good recovery data ranging from 89.00% to 111.33%.

An electrochemical biosensor for *Staphylococcus aureus* was designed based on a triple-helix molecular switch, controlling the switching of electrochemical signals [123].

Triple-helix DNA is a not standard nucleic acid structure, which inserts the third strand in the Watson–Crick double-strands structure [124]. The triple-helix DNA particular structure gives it unique properties and has been widely used in fluorescent and electrochemical biosensors. In addition, the triple-helix DNA has the same stability as the conventional double-stranded DNA [124]. When an aptamer sequence is introduced in the loop of triple-helix DNA, it has not to be reshaped and thus can detect various targets only by changing the aptamer. At the same time, the triple-helix DNA structure contains more nucleic acid strands, which can effectively improve the sensitivity of the biosensor [124]. In addition, triple-helix DNA has high flexibility. Through the special structure, the triple-helix DNA combined with specific antibodies can reversibly complete the load and release of the target sequence [124].

The biosensor showed a dynamic range from 30 to 3 × 10^8^ CFU mL^−1^, with a detection limit of 8 CFU mL^−1^. In addition, the sensor is used to detect *S. aureus* in spiked lake water, tap water and diluted honey samples, with acceptable recovery results. Using the particular biosensor design, the same sensing approach has also been successfully applied for *Escherichia coli* detection.

An impedimetric sensor based on bacteria-imprinted conductive poly(3-thiopheneacetic acid) (BICP) film was developed for the label-free detection of *S. aureus* [125]. The BICP film was in situ synthesized and deposited on the gold electrode surface. Many factors affecting the imprinting and recognition steps were studied and performed to obtain the optimal sensing performance.

Under the optimized conditions, a rapid recognition within 10 min, a limit of detection of 2 CFU mL^−1^, and a linear concentration range from 10 to 10^8^ CFU mL^−1^ were obtained. The sensor also showed high selectivity and repeatability.

Furthermore, the label-free impedimetric sensor was applied to the determination of *S. aureus* to artificially contaminated milk samples with good recovery results.

As a last and significant example, we introduce a dual signal amplification electrochemical biosensor based on a DNA walker and DNA nanoflowers to detect *S. aureus*, developed by the Zhou group [126] and illustrated in Figure 7.

Briefly, some details about the DNA walker are necessary. In addition to the construction of static DNA nanomaterials, DNA can also form molecular machines with dynamic behaviors [127]. Like the motor proteins responsible for the cellular movement, a DNA walker is a nanoscale molecular device/nanomachine driven by environmental stimulation, enzyme reaction, or strand displacement reaction [128,129]. It can carry out repeated mechanical cycle movement along the DNA orbit composed of nucleic acids to realize a signal cascade amplification [130].

Two groups of double-stranded DNA are immobilized on the surface of a gold electrode. The bond of *S. aureus* with its aptamer caused the disintegration of the long double-strands, so releasing the DNA walker. With the exonuclease III (Exo III) support, the DNA walker moves along the electrode surface, hydrolyzing the anchored short double-strands. After introducing a specially customized circular DNA and phi29 DNA polymerase, the rolling circle amplification (RCA) reaction was launched. DNA nanoflowers are formed at a high local concentration of DNA in the solution, creating binding sites for the electroactive label, i.e., methylene blue (MB) and thus yielding an intense signal. Under optimized conditions, the current response is linearly correlated to the logarithm of the *S. aureus* concentrations, ranging from 60 to 6 × 10^7^ CFU mL^−1^, and the detection limit is 9 CFU mL^−1^. Finally, the proposed biosensor has been applied to water samples and diluted honey samples, achieving good results in recovery.

#### 3.2.4. Listeria monocytogenes

*Listeria* genus consists of rod-shaped Gram-positive bacteria and includes seventeen different species [131]. Among them, *Listeria monocytogenes* (LM) is responsible for listeriosis in humans [131], and it was classified as an opportunistic, dangerous pathogen, especially for high-risk population groups, such as pregnant women, children, old, and immunosuppressed people. Listeriosis could lead to serious diseases, such as meningitis, fetal anomalies, abortion, febrile gastroenteritis or even generalized infection [131].

Despite the low incidence, when compared to other common foodborne diseases (e.g., *Salmonellosis* or *Escherichia coli* infections), LM infection is associated with a greater number of hospitalizations and a higher mortality rate (20–30%) [131]. In addition, the fact that LM can grow in different food commodities (i.e., dairy products, raw and preserved animal meats and vegetable products) [132] and in adverse environments exacerbates the problem.

Soni [133] provided a general overview concerning the emerging trends for *L. monocytogenes* detection, summarizing the developments in optical, piezoelectric, cell-based, and electrochemical biosensing detection applied in different fields, such as clinical diagnostics, food analysis, and environmental monitoring, and, also, evidencing their drawbacks and advantages.

Narang and coworkers published a review [134] concerning the evolution of analytical techniques for *L. monocytogenes*, highlighting the importance and the performances of the electrochemical ones.

Regarding the emerging electrochemical approaches for *L. monocytogenes* detection in food, Delerue-Matos [135] provided an accurate overview using low-cost electrochemical transducers, integration of new nanomaterials and incorporation of new bioreceptors in the sensing strategy.

Herein, we propose some significant and interesting approaches for LM detection in the electrochemical biosensing area.

The first *PlcA*-based nano-assembled electrochemical DNA biosensor [136] has been developed to detect *Listeria monocytogenes* in raw milk samples, using screen-printed carbon electrodes (SPCEs) modified with graphitized carbon nanofibers (CNFs) and gold nanoparticles (AuNPs).

Considering that the bacterium contains different virulent factors that disrupt the vacuolar membrane, the phosphatidylinositol-specific phospholipase C gene (PlcA) of *L. monocytogenes* is a virulent gene and encodes a 33-kDa protein responsible for the lysis of primary single-membraned vacuoles.

The selectivity of the developed biosensor was analyzed and confirmed using complementary and mismatch oligonucleotide sequences. The limit of detection was found to be 82 fg in 6 μL. The electrode was stable for six months. The validation study was performed using different milk samples artificially spiked with *L. monocytogenes,* and the results obtained demonstrated that it could be applied for *L. monocytogenes* detection in raw milk, moreover with good specificity.

The Gomes group [137] developed an innovative *Listeria* aptasensor using platinum interdigitated microelectrodes (Pt-IME). The sensor inserted in a particle/sediment trap has been used for the real-time analysis of irrigation water in a hydroponic lettuce crop. This system was used for rapid on-site analysis of water quality, using a smartphone-based potentiostat. In inflow conditions (100 mL samples), the aptasensor showed a detection limit of 48 CFU mL^−1^ with a linear range of 10^2^ to 10^4^ CFU mL^−1^. In no-flow conditions, the aptasensor was applied for *Listeria* detection to vegetable broth and hydroponic samples. Finally, this is the first example where an aptasensor has been used for testing microbial water quality for hydroponic lettuce in real time, using a smartphone-based acquisition system according to the standards. The aptasensor showed good recovery of 90%.

Delerue-Matos [138] reported the development and optimizing of an electrochemical immunosensor to detect LM p60 protein. A sandwich immunosensor using monoclonal and polyclonal antibodies for p60 protein secreted by *Listeria monocytogenes* and *Listeria* spp., respectively, were combined for assembling an efficient immunosensor.

To accomplish a more specific detection, different genes (hly, iap) and their corresponding encoded proteins, listeriolysin O and p60, considered as the major virulence factors associated with pathogenic action, have been targeted to detect *L. monocytogenes*.

Particularly, p60 protein has an important role in host cell invasion, cell division and viability, and besides being a cell surface protein, it is also secreted in large quantities into the growth media. These features make p60 an ideal diagnostic target for developing immunological detection systems.

A disposable screen-printed electrode was used as a transducer and monoclonal and polyclonal antibodies, specifically recognizing *Listeria monocytogenes* p60 protein and *Listeria* spp. p60 proteins were used as the sandwich immuno-pair, so the pathogenic *Listeria* can be distinguished from the nonpathogenic ones. The analytical signal was acquired through the voltammetric stripping of the enzymatically deposited silver, directly correlated to p60 concentration in the sample. In optimized conditions, a limit of detection of 1.5 ng mL^−1^ was obtained in less than 3 h. As a proof-of-concept, the proposed immunosensor was successfully applied to spiked milk samples, obtaining good results in recovery.

A novel electrochemical biosensor is reported for simultaneous detection of *Listeria monocytogenes* and *Staphylococcus aureus* [139]. The biosensor comprises gold nanoparticle-modified screen-printed carbon electrodes on which magnetic nanoparticles coupled to specific peptides were immobilized. Taking advantage of the proteolytic activities of the protease enzymes produced from the two bacteria on the specific peptides, the detection was achieved in 1 min. Limits of the detection of 9 CFU mL^−1^ for *Listeria monocytogenes* and 3 CFU mL^−1^ for *Staphylococcus aureus* were obtained. Good selectivity of the biosensor was demonstrated by analyzing samples containing at the same time *Staphylococcus aureus*, *Listeria monocytogenes* and *E. coli*. This platform seems to be promising for a rapid and cost-effective simultaneous detection of various bacteria. However, it should be noticed that an evaluation of the sensor analytical parameters on real samples, recovery included, was very useful, at least for one of the two pathogens.

As the last example, we introduce a non-electrochemical sensor, which integrates the sensitivity of magnetic sensing and efficiency of the hybridization reaction, providing an innovative and promising detection platform for pathogens.

A magnetic DNA sensor based on nucleic acid hybridization reaction and magnetic signal readout was proposed very recently by Chen [140]. This biosensing system allows the one-step detection of *L. monocytogenes* as low as 50 CFU mL^−1^ within 2 h without DNA amplification, and the average recovery in the spiked ham sausage samples resulted be 92.6%.

**Table 2 molecules-26-02940-t002:** Overview of recent electrochemical biosensors for pathogenic bacteria determination.

Electrode	(Bio)Sensor Format	Electrochemical Technique	Analyte/Sample	L.R.	LOD	References
GCE	Electrochemical immunosensor based on high-density gold nanoparticles (AuNPs), dispersed in chitosan (CHI) hydrogel, and modified glassy carbon electrode (GCE)	DPV	*Salmonella*/milk, water	10–10^5^ CFU mL^−1^	5 CFU mL^−1^	[100]
SPCEs	Label-free impedimetric aptasensor assembled by grafting a diazonium-supporting layer onto screen-printed carbon electrodes (SPCEs), followed by chemical immobilization of aminated-aptamer	EIS	*Salmonella*/apple juice	10–10^8^ CFU mL^−1^	6 CFU mL^−1^	[101]
AuE	Label-free impedimetric aptasensor based on combining poly-[pyrrole-co-3-carboxyl-pyrrole] copolymer and the *Salmonella* aptamer	EIS	*Salmonella*/apple juice	10^2^–10^8^ CFU mL^−1^	3 CFU mL^−1^	[102]
GCE	Electrochemical aptasensor developed using electrochemically reduced graphene-oxide–chitosan (rGO–CHI) composite deposited onto GCE	DPV	*Salmonella*/chicken	10–10^6^ CFU mL^−1^	10 CFU mL^−1^	[103]
AuE	Electrochemical aptasensor developed by combining target-induced aptamer displacement on gold nanoparticles (AuNPs) deposited onto Au electrode with rolling circle amplification (RCA)	DPV	*Salmonella*/milk, mineral water	20–20^7^ CFU mL^−1^	16 CFU mL^−1^	[104]
GF-GCE	Electrochemical immunosensor based on anti- *Salmonella* antibody immobilized on the surface of the graphite felt electrode	OSWV	*Salmonella*/no real samples	-	10^5^ *E. coli* cells mL^−1^	[105]
BiSPCE	Immunosensor where bacterial cells were separated immunomagnetically and reacted with conjugate; labeled with an electrochemical indicator, including hyperbranched dendron molecules and heavy metal-derived quantum dots (CdTe QDs). Square-wave anodic stripping voltammetry (SWASV) employing screen-printed carbon electrodes with in situ formed Bi(III) film (BiSPCE) was used for the detection and quantification of metal ions released from the QDs and correlated with the bacterium amount	SWASV	*Salmonella*/milk	-	4 CFU mL^−1^	[106]
AuIME	Electrochemical aptasensor using aptamer-coated gold-interdigitated microelectrode (IAuE) for target capture and impedance measurement, and antibody-modified nickel nanowires (NiNWs) for target separation and impedance amplification	EIS	*Salmonella*/chicken	10^2^–10^6^ CFU mL^−1^	80 CFU mL^−1^	[107]
AuIME	Immunosensor using multiple magnetic nanobead (MNB) nets in a ring channel for continuous-flow separation of target bacteria from the sample volume, manganese dioxide nanoflowers (MnO_2_ NFs) for efficient amplification of the biological signal, and an interdigitated microelectrode to measure impedance change	EIS	*Salmonella*/chicken	30–30 × 10^5^ CFU mL^−1^	19 CFU mL^−1^	[108]
AuIME	Impedimetric immunosensor using rotary magnetic separation and cascade reaction	EIS	*Salmonella*/chicken	10–10^6^ CFU mL^−1^	10 CFU mL^−1^	[109]
AuE	Electrochemical genosensor based on the immobilization of complementary DNA on the gold electrode surface, which hybridizes with a pathogen-specific fragment gene to make a sandwich structure	DPV	*E. coli*/beef	-	1.97 × 10^−^^14^ M	[113]
Magnetic-graphite epoxy composite (m-GEC) electrode (m-GECE)	Electrochemical magneto-genosensor based on the detection of the tagged amplified DNA obtained by single-tagging PCR with a set of pathogen-specific primers, followed by electrochemical magneto-genosensing on silica magnetic particles	Amperometry	*E. coli*/no real samples	0.03–3 ng mL^−1^	0.05 ng mL^−^^1^	[114]
AuE	Label-free impedimetric immunosensor using reduced graphene wrapped copper (II)-assisted cysteine hierarchical structure (rGO–CysCu) as the sensing layer	EIS	*E. coli*/water, fruit juice, milk	10–10^8^ CFU mL^−1^	3.8 CFU mL^−1^	[115]
GCE	ECL aptasensor based on AgBr nanoparticles (NPs) anchored on 3D nitrogen-doped graphene hydrogel (3DNGH) nanocomposites for immobilizing luminol and enhancing its ECL behavior	ECL	*E. coli*/meal samples	0.5–500 CFU mL^−1^	0.17 CFU mL^−1^	[116]
SPCEs	Label-free impedimetric aptasensor using 3D hierarchical nanostructured bridged rebar graphene (BRG) for modifying SPCES	EIS	*E. coli*/water, juice, and milk.	10^2^–10^6^ CFU mL^−1^	10 CFU mL^−1^	[117]
PGE	Electrochemical immunosensor based on the PPy/AuNP/MWCNT/CHI hybrid nanocomposite modified pencil graphite electrode (PGE)	Amperometry	*E. coli*/no real samples	30–30^6^ CFU mL^−1^	30 CFU mL^−1^	[118]
SPCEs	Electrochemical immunoassay using silica-coated Fe3O4 magnetic nanoparticles (Fe3O4@SiO2) and Au@Pt nanoparticles loaded on neutral red (NR) functionalized graphene to form composite complex rGO–NR–Au@Pt	CV	*E. coli*/pork and milk	4.0 × 10^3^–4.0 × 10^8^ CFU mL^−1^	4.0 × 10^2^ CFU mL^−1^	[119]
GF-GCE	Electrochemical immunosensor based on anti- *Escherichia coli* antibody immobilized on the surface of the graphite felt electrode	OSWV	*E. coli*/beef	-	400 cells mL^−1^	[120]
AuIME	Electrochemical biosensor based on hybridization chain reaction (HCR)	MSPQC	*S. aureus*/milk and human serum	50–10^7^ CFU mL^−1^	50 CFU mL^−1^	[122]
AuE	Electrochemical biosensor based on a triple-helix molecular switch, which can control the switching of electrochemical signals	DPV	*S. aureus*/water and honey	30–30 × 10^8^ CFU mL^−1^	8 CFU mL^−1^	[123]
AuE	Label-free impedimetric immunosensor based on bacteria-imprinted conductive poly(3-thiopheneacetic acid) (BICP) film	EIS	*S. aureus*/milk	10–10 × 10^8^ CFU mL^−1^	2 CFU mL^−1^	[125]
AuE	Dual signal amplification electrochemical biosensor based on a DNA walker and DNA nanoflowers	DPV	*S. aureus*/water and honey	60–60 × 10^7^ CFU mL^−1^	9 CFU mL^−1^	[126]
SPCNF/AuNPsE	*plcA*-based electrochemical DNA biosensor using screen-printed CNF/AuNPs electrode	CV	*L. monocytogenes*/milk	0–0.234 ng/6 μL	82 fg/6 μL	[136]
Pt-IME	Aptasensor using platinum interdigitated microelectrodes (Pt-IME) biofunctionalized with *Listeria*-specific aptamer and a smartphone-based signal acquisition system	EIS	*L. monocytogenes*/vegetable broth, hydroponic media	10^2^–10^6^ CFU mL^−1^	23 CFU mL^−1^	[137]
SPCEs	Electrochemical immunosensor using a disposable screen-printed electrode as transducer surface and monoclonal and polyclonal antibodies specifically recognizing *Listeria monocytogenes* p60 protein used as the sandwich immuno-pair	CV	*L. monocytogenes*/milk	5–150 ng mL^−1^	1.5 ng mL^−1^	[138]
Disposable electrical printed (DEP) microarray electrode s	Electrochemical biosensor assembled by selectively functionalizing the array electrodes with bacteria-specific peptides	SWV	*L. monocytogenes*/no real samples	10–10^7^ CFU mL^−1^	9 CFU mL^−1^	[139]

Abbreviations: AuE: gold electrode; AuNPs: gold nanoparticles; AuSPE: gold screen-printed electrode; Au-IME: gold interdigitated microelectrodes; CA: chronoamperometry; CF: carbon felt; CNF: carbon nanofibers; CV: cyclic voltammetry; DEP: Disposable electrical printed microarray electrode; DPV: differential pulse voltammetry; EIS: electrochemical impedance spectroscopy; GCE: glassy carbon electrode; GF: graphite felt; GO: graphene oxide; ITO: indium–tin-oxide electrode; MBs: magnetic beads; m-GEC: magnetic graphite-epoxy composite; MWCNTs: multi-walled carbon nanotubes; OSWV: Osteryoung square-wave voltammetry; PPY: polypyrrole; QDs: quantum dots; Pt-IME: platinum interdigitated microelectrodes; SPCE: screen-printed carbon electrode; SWV: square-wave voltammetry; SWASV: square-wave anodic stripping voltammetry.

### 3.3. Pesticides

Pesticides are among the most used products in the agri-food industry for the control, prevention, and elimination of pests. According to the target pest, they can be classified as insecticides, fungicides, herbicides, etc. The main classes of pesticides are the following carbamates, organophosphates, pyrethroids, or triazines, among others [141] and all these compounds resulted highly toxic. According to the World Health Organization (WHO), they can be classified as carcinogenic, neurotoxic, or teratogenic [142].

The maximum residual limits (MRLs) legally permitted in the European Union are 0.1 μg/L for a single pesticide and 0.5 μg/L for total pesticides [143].

Therefore, for pesticide monitoring at the required MRLs, many electrochemical biosensors using different analyzing techniques were developed, and some relevant examples are shown in Table 3.

#### 3.3.1. Insecticides

Mukherjee [144] presented a general overview of the recent advancements concerning different acetylcholinesterase (AChE) inhibition-based sensing strategies, including optical, electrochemical, lab-on-paper sensors, microfluidic, and other devices for the rapid detection of organophosphorus (OPs) pesticides, developed in the last two years.

Kumar [145] focused his recent review on the biosensors developed in the last thirty years to detect a particular organophosphate insecticide: dichlorvos, widely used in agriculture and industry. His review described the progressive development of biosensors from using conventional immobilizing supports to more advanced hybrid/composite nanomaterials, also summarizing developing biosensors by enzyme inhibition methods.

A glassy carbon electrode (GCE) modified with single-walled carbon nanohorns (SWCNH) and zein (ZE), a prolamin type-protein find in maize, was proposed by Janegitz [146] for the fenitrothion (FT) determination using differential pulse adsorptive cathodic stripping voltammetry (DPACSV).

Fenitrothion (FT) is an organophosphorus pesticide with cholinesterase inhibitory action, and it is widely used for insect control in grains and in vegetable cultures.

The sensor showed a linear response ranging from 9.9 × 10^−7^ to 1.2 × 10^−5^ mol L^−1^, with a limit of detection of 1.2 × 10^−8^ mol L^−1^. The proposed sensor was successfully applied to determine FT pesticide in spiked natural water and orange juice samples. Moreover, the electrochemical sensor showed good repeatability and reproducibility.

A cellulose microfiber supported reduced graphene-oxide composite was employed for modifying a screen-printed carbon electrode (SPCE) for determining fenitrothion (FT) in water samples using differential pulse voltammetry (DPV) [147].

A linear concentration range up to 1.134 mM with a detection limit of 8 nM was obtained. To validate the sensor, it was applied to detect fenitrothion in different spiked water samples, obtaining acceptable recovery results.

A biosensor for determining methyl parathion (MP, organophosphorus pesticide) using glutaraldehyde (Glu) crosslinked with acetylcholinesterase (AChE) immobilized on single-wall carbon nanotubes (SWCNTs) enveloped with bovine serum albumin (BSA) was realized by Sundramoorthy [148].

The proposed biosensor exhibited a linear range from 1 × 10^−10^ M to 5 × 10^−6^ M with a limit of detection of 3.75 × 10^−11^ M and showed good repeatability and reproducibility. In addition, it was applied to real samples, such as spiked strawberry and apple juices, obtaining good results in terms of recovery.

Poly-3,4-ethylenedioxythiophene (PEDOT) membrane and zirconia nanoparticles (ZrO_2_ NPs) were directly synthesized on ITO electrode and successively employed for methyl parathion (MP) electrochemical detection [149]. Combining the individual properties of PEDOT (conductivity and electrocatalysis) and of ZrO_2_ NPs (affinity to MP), the resulting sensor showed a limit of detection of 2.8 ng·mL^−1^ and a linear concentration range of 5–2000 ng·mL^−1^. Furthermore, this sensor exhibited acceptable selectivity and reproducibility. The sensor was applied to spiked water samples with acceptable recovery results.

Using ultrathin MXene nanosheets (i.e., two-dimensional (2D) transition metal carbides and nitrides) as a natural reducing agent and support, the shape-controlled Au–Pd bimetallic nanoparticles via a self-reduction process were synthesized for enhancing the performance of the resulted biosensor and supporting the acetylcholinesterase immobilization. Using this multidimensional nanocomposite (MXene/Au–Pd) as a functional platform, a disposable electrochemical biosensor to detect paraoxon, an organophosphorus pesticide, was developed [150], as illustrated in Figure 8.

Under the optimized conditions, this biosensor showed a linear concentration range from 0.1 to 1000 μg L^−1^, with a detection limit of 1.75 ng L^−1^. Furthermore, the biosensor was applied for paraoxon detection in spiked pear and cucumber samples with promising results.

A smart bioelectrode based on redox-active protein hemoglobin (Hb) has been prepared to determine methyl parathion (MP) [151]. The bioelectrode has been designed by immobilizing Hb on electrochemically reduced graphene-oxide–chitosan-based biocompatible coatings. The sensor showed a detection limit of 79.77 nM with good reproducibility. The biosensor also was applied to spiked vegetable samples with interesting recovery results, ranging from 94% to 101%.

Another nanocomposite was synthesized via reducing graphene oxide on fumed silica (FS) surface to develop a modified electrode to determine fenitrothion (FT). Reduced graphene oxide (RGO) decorated with FS (FS@RGO) was used to modify the glassy carbon electrode (GCE) [152]. FS on nanocomposite allowed a homogeneous distribution of the nanocomposite. Moreover, the presence of FS brings additional functionality to the FS@RGO nanocomposite, which increases the adsorption and electron transfer rate of FNT. The FS@RGO/GC electrode showed a linear concentration range from 0.005 to 1.0 μM and a limit of detection of 0.00019 μM. The performance of the FS@RGO/GC electrode was evaluated using recovery studies in river water, urine, and in different fruit and vegetable extracts (raisin, tomato, and orange), and acceptable recovery values between 92.3% and 112.2% were obtained.

An electrochemical sensor, using a glassy carbon electrode modified with a dodecane film where silver nanoparticles have been electrodeposited, was developed for fenitrothion detection [153]. The glassy carbon electrode was coated with a dodecane film by drop-casting technique. Silver nanoparticles were electrodeposited on the dodecane layer, and the resulted modified GCE has been used for electrochemical fenitrothion detection.

The electrode was found to be stable with constant sensitivity for many cycles of the analysis. The results observed in the electrochemical approach were similar to those obtained by the HPLC technique. Finally, it was applied to spiked vegetable samples, such as potatoes and paddy grains.

A *Tribolium castaneum* (red flour beetle) acetylcholinesterase (Tc-AChE)-based electrochemical biosensor integrating WO_3_/g-C_3_N_4_ nanocomposite modified pencil graphite electrode was developed to detect an organophosphate insecticide, Phosmet [154].

Graphitic carbon nitride g-C_3_N_4_ alone does not possess good electrical conductance, which can be improved by doping or coupling with other nanomaterials, such as tungsten trioxide (WO_3_).

The WO_3_/g-C_3_N_4_ nanocomposite provides a nontoxic, biocompatible surface for immobilizing the enzyme, providing a large surface area, high conductivity, and low ohmic resistance. The proposed biosensor showed a good analytical performance with a low detection limit of 3.6 nM for Phosmet. The biosensor was also applied to detect Phosmet in spiked wheat samples with a 99% recovery rate.

In the next example, graphitic carbon nitride g-C_3_N_4_ was modified to improve its electrical conductivity by coupling with another nanomaterial, such as strontium hexaferrite (SrFe_12_O_19_) nanorods. The resulting nanocomposite was used to modify screen-printed carbon electrodes (SPCEs) to detect fenitrothion [155]. The resulted electrochemical sensor performed a good detection range from 0.005 to 378.15 μM with a low detection limit of 0.0014 μM. The SrFe_12_O_19_/g-C_3_N_4_-modified sensor was applied to determine FTN to different spiked fruit samples with acceptable recoveries.

#### 3.3.2. Herbicides

A recent review [156] provided an overview concerning the development, applicability, and performances of nanomaterials-based immunosensors for the pesticides and herbicides detection in water, food, and soil samples.

The use of nanomaterials for the immunosensing system assembling was found to be a smart option to implement an effective and selective sensing platform for pesticides/herbicides analysis, combining different nanomaterials, such as graphene, carbon nanotubes, metal nanoparticles, etc. with different sensing methodologies (e.g., electrochemical, optical, and quartz crystal microbalance (QCM)).

A recent review [157] about the determination of paraquat (PQ) in foods, including milk, apple, tomato juices, and potato samples, using electrochemical methods combined with several modified electrodes was reported by Mhammedi. Paraquat is widely used as an herbicide (broadleaf weed killer), owing to its excellent effect for crop protection and horticultural use, but it resulted very toxic, and its detection, possibly on-site, is required.

The importance of the electrode modifiers combined with the most suitable electrochemical sensing technique was underlined.

A very particular and ecofriendly method [158] was developed to determine trifluralin. Trifluralin is an herbicide affecting endocrine function, and so it is listed as an endocrine disruptor in the European Union list [159].

The trifluralin sensor is based on its electrochemical oxidation on a three-electrode system designed directly on the surface of an agricultural product, using Ag-citrate/GQDs (graphene quantum dots) nano-ink. The sensor was prepared by writing directly on the surface of the samples, obtaining, for example, Ag-citrate/GQDs nano-ink/leaf electrode if testing trifluralin in an apple.

Under optimized experimental conditions, this sensor was exhibited good sensitivity and specificity for trifluralin detection. The obtained linear range was between 0.008 to 1 mM, and the limit of quantification was 0.008 mM, using cyclic voltammetry. In addition, the obtained linear range using differential pulse voltammetry (DPV) and square wave voltammetry (SWV) is 0.005–0.04 mM with the limit of quantification of 0.005 mM. For further validation of the applicability of the proposed method, it was also used to detect trifluralin on the surface of apple skin. In Figure 9, after assembling the electrode, its electrochemical behavior was evaluated in the absence and presence of trifluralin using CV, DPV and SWV techniques. CVs were performed using the Ag-citrate/GQDs nano-ink/leaf electrode at a potential of −1.0 to +1.0 V and the scan rate of 100 mV/s. As evident in Figure 9A, the oxidation peak appeared in the presence of trifluralin at 0.4 V. On the contrary, in the PBS (as blank), no electrochemical behavior was observed. In addition, the results obtained from the more sensitive DPV and SWV techniques confirm the conductivity of the sensor provided and its ability to detect the analyte. It was hypothesized [158] that the interaction mechanism of trifluralin on the prepared Ag-citrate/GQDs nano-ink/leaf electrode was based on electrostatic interactions between the negatively charged dipole nitro group of trifluralin and the protonated amine group of chitosan, also involving Ag^+^ ions of Ag-citrate present in the conductive nano-ink.

A study concerning miniaturized lab-on-a-chip platforms for online analysis of the pesticide-nucleic acid interactions has been reported by Congur [160].

Glyphosate (GLY) is a broad-spectrum herbicide used worldwide to control grass weeds. Although it was evaluated as a non-toxic agent in 20th-century, its carcinogenic and genotoxic effects have been intensively investigated in the last decade. Moreover, combining GLY and 2,4-dichlorophenoxyacetic acid (2,4-D) has been widely applied as an herbicide mixture. Although genotoxicity of GLY has been evaluated in vivo studies, there is no report in the literature for the in vitro biointeraction monitoring of GLY, and double-stranded DNA, or how combining GLY and 2,4-D affects DNA. For this reason, an electrochemical biosensor platform was developed for investigating the pesticide–DNA interaction by using disposable pencil graphite electrodes (PGEs). First, a voltammetric investigation of the interaction between GLY and DNA was carried out. In addition, the combined genotoxic effects of the mixture of GLY and 2,4-D or the mixture of their herbicide forms onto DNA were monitored. This effect was concentration-dependent. In addition, it was evidenced that GLY as herbicide or the mixture of herbicides of GLY and 2,4-D had more genotoxic effect than the analytical grade of GLY and 2,4-D. Finally, these disposable PGEs provide a robust, ecofriendly sensing platform for monitoring herbicide–DNA interaction with sensitive and reliable results.

#### 3.3.3. Fungicides

A general and recent review summarized [161] the current analytical approaches and techniques used to analyze dithiocarbamate fungicides (DTFs) widely used to control fungal diffusion in crops and ornamental plants.

It included chromatography, spectroscopy, and sensor-based approaches and discussed the challenges related to selectivity, sensitivity, and sample preparation. Finally, biosensors based on enzymatic inhibition are considered very promising as analytical tools for DTFs.

A simple, inexpensive, sensitive, selective electrochemical approach was developed for simultaneous quantification of the fungicides thiram and carbendazim [162] in samples of honey, fresh grape juice, and in agricultural formulation using a carbon paste electrode modified with zeolite.

For thiram, a linear concentration range of 0.36–4.99 × 10^−7^ mol L^−1^, with a limit of detection of 6.74 × 10^−9^ mol L^−1^. For carbendazim, a linear concentration range of 0.10–2.35 × 10^−6^ mol L^−1^, with a limit of detection of 3.51 × 10^−9^ mol L^−1^.

Thiram and carbendazim recovery experiments were performed on spiked honey and grape juice samples, yielding recovery rates in the range of 98.85–101.15%. In the agricultural formulation, the concentrations measured with the new method were close to those specified on the label, with deviations below 1.1%. No thiram or carbendazim was found in the grape juice and honey samples. The results demonstrated the sensor applicability for quantifying both compounds simultaneously in real samples.

Nanoporous gold (NPG) with unique structural and functional properties was selected as a recognition key element of an electrochemical sensor for the simultaneous detection of methyl parathion (MP) and carbendazim (CBM) [163].

As a recognition element affecting sensitivity and selectivity, the modified material of the electrode is a key factor for electrochemical sensor construction in pesticide determination. Recently, nanoporous gold (NPG) with unique properties has received increasing interest and attention in electrochemical catalysis and electrochemical sensor construction [164]. NPG can be easily fabricated by dealloying gold alloys that possess a continuous open structure, large surface area, high conductivity, and strong binding ability to the electrode. As an electrocatalyst, NPG offers some advantageous properties: (1) it can be easily prepared, recovered, and recycled; (2) its surface structural properties allow further surface functionalization; (3) its porous structure and film properties (approximately 100 nm thickness) make it easy to be integrated into device platforms.

Because of its geometrical features, such as irregularity, large roughness, and high porosity, which allow more active sites on the electrode surface and increased probability for charge transfer between molecules and electrode surface and the resulting nanostructure, NPG was turned out to be capable of sensing without enzyme by directly oxidizing small molecules, such as dopamine, ascorbic acid, hydrazine, and several environmental pollutants.

Due to the good performance of the NPG-based sensor reported previously, NPG can represent an ideal electrode material for the simultaneous determination of pesticides, such as MP and CBM.

For the detection of MP and CBM, good linear responses were observed in a concentration ranges of 0.5–150 μM for MP and 3.0–120 μM for CBM, with low detection limits of 0.02 μM for MP and 0.24 μM for CBM. Additionally, the NPG/GCE electrode presented good specificity, selectivity, and it was applied for detecting the two fungicides in water samples, with interesting results in terms of recovery.

Gadolinium-oxide nanorods embedded on the graphene aerogel (GdO NRs/GA) were employed for assembling a selective electrochemical sensor to detect carbendazim (CBM) [165].

The GdO NRs/GA-modified electrode showed good analytical performances. Interestingly, the GdO NRs are strongly anchored in the GA matrix, providing an efficient pathway for rapid electron transfer. A linear concentration range from 0.01 to 75 μM with a low detection limit of 3.0 nM was achieved. The sensor was applied to the spiked water sample, and the results are comparable with those obtained from the HPLC technique, with a recovery ranging from 97.80−99.40%.

An interesting example of on-site pesticide monitoring in food is reported by Raymundo-Pereira [166]. It is a nonenzymatic electrochemical sensor, but the approach seems to be particularly innovative and, on my opinion, worthy of being mentioned and highlighted.

A set of three glove-embedded sensors printed on three fingers of a rubber glove allowed the selective, sensitive, and simultaneous detection of different pesticides, such as carbendazim (carbamate), diuron (phenylamide), paraquat (bipyridinium) and fenitrothion (organophosphate). Figure 10 illustrates the design and working principle of the glove-embedded sensors.

The sensors consisted of a pretreated screen-printed carbon electrode and two other such electrodes coated with either carbon spherical shells (CSS) or Printex carbon nanoballs (PCNB). Detection of carbendazim and diuron was performed using differential pulse voltammetry (DPV) and the electrodes coated with CSS and PCNB, respectively, with limits of the detection of 4.7 × 10^−8^ and 9.2 × 10^−7^ mol L^−1^. Square wave voltammetry (SWV) was applied to detect paraquat and fenitrothion with limits of the detection 2.4 × 10^−8^ and 6.4 × 10^−7^ mol L^−1^ using the pretreated electrode in sulfuric acid solution. To investigate the applicability of the sensors to real food samples, spiked samples of cabbages and apples and orange juice were analyzed. The finger contacted the food sample during the analysis.

The recoveries varied between 90 and 110%, indicating that the glove-based sensors are selective and effective to detect carbendazim, diuron, paraquat and fenitrothion in real food samples. The interference from other pesticides, such as chlorpyrifos, carbaryl, methomyl, atrazine, trifluralin, glyphosate and chloranil, was investigated and resulted negligible from the experimental evidence.

**Table 3 molecules-26-02940-t003:** An overview of recent electrochemical biosensors for pesticide determination.

Electrode	(Bio)Sensor Format	Electrochemical Technique	Analyte/Sample	L.R.	LOD	References
GCE	Electrochemical sensor using GCE modified with MCNHs and zein (SWCNH-ZE/GCE)	DPACSV	Fenitrothion/orange juice	9.9 × 10^−7^–1.2 × 10^−5^ M	1.2 × 10^−8^ M	[146]
SPCEs	Electrochemical sensor based on SCPCEs modified with cellulose microfibers supported reduced graphene-oxide composite	DPV	Fenitrothion/water	0.03–1333.8 μM	8 nM	[147]
GCE	Electrochemical biosensor using glutaraldehyde (Glu) crosslinked with acetylcholinesterase (AChE) immobilized on s-SWCNTs wrapped with bovine serum albumin (BSA)	DPV	Parathion/strawberry and apple juice	1 × 10^−10^–5 × 10^−6^ M	3.75 × 10^−^^11^ M	[148]
ITO	Electrochemical sensor using ITO electrode modified with poly-3,4-ethylenedioxythiophene (PEDOT) membrane and zirconia nanoparticles (ZrO_2_ NPs)	CV	Parathion/water	5–2000 ng·mL^−1^	2.8 ng·mL^−^^1^	[149]
SPCEs	Electrochemical biosensor using the multidimensional nanocomposite (MXene/Au–Pd) as the functional platform for immobilizing AChE	Amperometry	Paraoxon/pear and cucumber	0.1–1000 μg L^−1^,	1.75 ngL^−1^	[150]
Fluorine–tin-oxide glass electrodes (FTO)	Electrochemical sensor developed by immobilizing hemoglobin (Hb), redox-active proteins on electrochemically reduced graphene-oxide–chitosan (ERGO–CHI/Hb/FTO)	SWV	Parathion/onion, lettuce	0.076–0.988 mM	79.77 nM	[151]
GCE	Electrochemical sensor using reduced graphene oxide (RGO) decorated fumed silica (FS) to modify glassy carbon (FS@RGO–GCE)	DPV	Fenitrothion/orange juice and tomato	0.005–1.0 μM	0.00019 μM	[152]
GCE	Electrochemical sensor using silver nanoparticles/dodecane modified glassy carbon electrode	DPV	Fenitrothion/paddy grains and potato	0.1–7 nM	0.60 nM	[153]
PGE	Electrochemical biosensor using WO_3_/g-C_3_N_4_ nanocomposite modified pencil graphite electrode as an immobilizing platform for *Tribolium castaneum* (red flour beetle) acetylcholinesterase (Tc-AChE)	Amperometry	Phosmet/wheat flour	5–125 nM	3.6 nM	[154]
SPCEs	Electrochemical sensor based on strontium hexaferrite (nanorods) decorated on porous graphitic carbon nitride (SrFe_12_O_19_/g-C_3_N_4_) to modify SPCEs	DPV	Fenitrothion/grapes, apricots, orange, cranberry, guava, mango	0.005–378.15 mM	1.4 nM	[155]
Ag–citrate/GQDs nano-ink/leaf or skin	Electrochemical sensor prepared by direct writing on the surface of the samples, using Ag-citrate/graphene quantum dots (GQDs) nano-ink	DPV, SWV	Trifluralin/apple skin	0.005–0.04 mM	0.005 mM	[158]
PGE	Electrochemical biosensor platform developed to detect the pesticide–DNA interaction by using disposable pencil graphite electrodes (PGEs) where DNA was immobilized via passive absorption	DPV	Monitoring glyphosate and 2,4-dichlorophenoxyacetic acid DNA interactions	-	-	[160]
CPE	Electrochemical sensor using a carbon paste electrode modified with recrystallized zeolite	SWV	Thiram and carbendazim/honey and grape juice	0.36–4.99 × 10^−7^ M Thiram 0.10–2.35 × 10^−^^6^ M Carbendazim	6.74 × 10^−^^9^ M Thiram 3.51 × 10^−9^ M Carbendazim	[162]
NPG–GCe	Electrochemical sensor based on modified GCE with nanoporous gold film	DPV	Carbendazim and methyl parathion/water	0.5–150 mM Methyl parathion 3.0–120 mM Carbendazim	0.02 mM methyl parathion 0.24 mM Carbendazim	[163]
GCE	Electrochemical sensor based on gadolinium-oxide nanorods embedded on the graphene aerogel (GdO NRs/GA)	CV	Carbendazim/water	0.01–75 μM	3.0 nM	[165]
SPCEs	Electrochemical sensor based on SPCEs modified with carbon spherical shells (CSS) or Printex carbon nanoballs (PCNB)	DPV	Carbendazim and diuron/cabbages, apples, and orange juice	0.1–1.0 μM carbendazim 1–10 mM diuron	4.7 × 10^−^^8^ M carbendazim 9.2 × 10^−^^7^ M diuron	[166]
SPCEs	Electrochemical sensor based on SPCEs modified with carbon spherical shells (CSS) or Printex carbon nanoballs (PCNB)	SWV	Paraquat and fenitrothion/cabbages, apples, and orange juice	0.1–1.0 μM paraquat 1–10 mM fenitrothion	2.4 × 10^−^^8^ M paraquat 6.4 × 10^−^^7^ M fenitrothion	[166]

Abbreviations: AuE: gold electrode; AuNPs: gold nanoparticles; AuSPE: gold screen-printed electrode; CA: chronoamperometry; CPE: carbon paste electrode; CSS: carbon spherical shells; CV: cyclic voltammetry; DPACSV: differential pulse adsorptive cathodic stripping voltammetry DPV: differential pulse voltammetry; EIS: electrochemical impedance spectroscopy; GQDS: graphene quantum dots; GCE: glassy carbon electrode GO: graphene oxide; ITO: indium–tin-oxide electrode; MWCNTs: multi-walled carbon nanotubes; NPG: nanoporous gold; QDs: quantum dots; PCNB: Printex carbon nanoballs; PGE: pencil graphite electrode; SPCE: screen-printed carbon electrode; SWV: square-wave voltammetry; SWASV: square-wave anodic stripping voltammetry.

### 3.4. Antibiotics

Antibiotics are a group of pharmaceutical drugs widely used in human and veterinary medicine for treating many different infectious diseases [167].

Large quantities of antibiotics are used annually in the livestock industry and aquaculture worldwide, and antibiotic use for animals can produce antibiotic residues in food products, such as meat, chicken, egg, milk, honey, and fish [168].

Residues of these drugs can induce several toxic effects in humans [169]. The most common side effect of antibiotic residues in foodstuffs is developing antimicrobial resistance. The resistant bacterial pathogens can be transferred to humans through the food chain and cause the inefficiency of antibiotic therapy [168,169].

To minimize the adverse effects of antibiotics, the European Union has banned some specific antimicrobial, while for those not banned, maximum residue limits (MRLs) have been established to ensure consumer safety from ingestion of antibiotic residue in animal-derived foods [170].

To guarantee that the residue of antibiotics in animal-derived foods is below the MRL, it is very important to find suitable methods to determine the content of antibiotics.

Several reviews reported the biosensing approaches for antibiotics detection. In particular, Marty presented [171] and highlighted the achievements in developing biosensors in the above-mentioned application field, evidencing the different types of involved nanomaterials and the biorecognition elements.

Very recently, Liang [172] revised the current antibiotic detection technologies, including chromatography, mass spectrometry, capillary electrophoresis, optical detection, and electrochemistry and evidencing the advantages and drawbacks.

Herein, we reported the most recent advances in electrochemical biosensors to detect antibiotics in food and Table 4 summarizes the analytical characteristics of the electrochemical biosensors for antibiotics reported in the review.

A multiplexed electrochemical aptasensor for multiple antibiotics detection, using kanamycin (KANA) (MLR 150 mg/kg in milk [170]) and ampicillin (AMP) (MLR 4 mg/kg in milk [170]) as model analytes, was assembled by using metal ions encoded apoferritin probes and double stirring bars-assisted target recycling for signal amplification [173].

KANA and AMP were determined simultaneously within the range from 0.05 pM to 50 nM, and the detection limits were 18 fM KANA and 15 fM AMP. The feasibility of the aptasensor was evaluated by testing milk and fish samples. Spiked samples with KANA and AMP were employed, and the analytical results were coherent and comparable with those obtained by ELISA.

The sensing approach of an electrochemical aptasensor [174] to detect ampicillin (AMP) is based on applying a ladder-shaped DNA structure as a multilayer physical block on the surface of the gold electrode. Sensitive detection of AMP was obtained with a detection limit of 1 pM, probably due to the electrostatic repulsion and physical hindrance of the ladder-shaped DNA structure.

The aptasensor response was linear in the concentration range from 7 pM to 100 nM. The aptasensor was applied to spiked milk samples with satisfactory results.

He [175] developed a disposable and portable aptasensor for the fast and sensitive detection of oxytetracycline (OTC) (MLR 100 mg/kg in muscle [170]) using gold nanoparticles (AuNPs)/carboxylated multi-walled carbon nanotubes (cMWCNTs)@thionine connecting complementary strand of aptamer (cDNA) as signal tags. The substrate electrode of the aptasensor was a portable thin-film gold electrode (TFGE).

In the presence of OTC, OTC competed with cDNA to combine with aptamer. The bioconjugate (AuNPs/cMWCNTs/cDNA@thionine) was released from the TFGE, and the electrochemical signal decreased.

Under optimized conditions, the aptasensor showed a dynamic range of 1 × 10^−13^–1 × 10^−5^ g mL^−1^ for OTC with a low detection limit of 3.1 × 10^−14^ g mL^−1^ and was applied to determine OTC to spiked chicken samples with satisfactory performances.

Another aptasensor, based on the protective effect of aptamer-antibiotic conjugate towards endonuclease DpnII activity, was developed [176] for determining ampicillin in milk and water samples.

Regarding the details, without ampicillin, DNA aptamer first hybridizes with the capture probe to form double-strand DNA (dsDNA) structure. Then, dsDNA is cut by DpnII restriction endonuclease to form two dsDNA fragments. One fragment is released from the electrode surface, and the other fragment is kept on the electrode surface. Then, the dsDNA fragment kept on the electrode surface is further cut by exonuclease III (Exo III), so causing the dsDNA fragment to release from the electrode surface. Thus, the electrochemical signal increases due to the decrease of the interface electron transfer resistance due to the release of dsDNA from the electrode surface. However, forming dsDNA is hindered by forming aptamer-ampicillin conjugate and preventing the digestion of DpnII and Exo III towards the capture probe. Thus, a weak electrochemical signal is obtained, resulting in increased interface electron transfer resistance due to the presence of the dsDNA fragments on the electrode surface. Following the relationship between ampicillin concentration and the decrease of the electrochemical signal, ampicillin is detected with a low detection limit of 32 pM, which is lower than the MRL allowed by the European Union (5 μg kg^−1^)). The developed method also presents good selectivity. Moreover, the applicability is tested by detecting antibiotics in spiked milk and water samples with satisfactory results.

A novel “signal-on” sensing strategy for sensitive electrochemical determination of tetracycline (TC) (MLR 100 mg/kg in milk [170]) was reported [177] for the first time, including elements of triple-helix aptamer probes (TAP), catalyzed hairpin assembly (CHA) signal amplification and host–guest recognition. Under optimal conditions, a linear relation along with the logarithm of the TC concentrations ranging from 0.2 nM to 100 nM and a detection limit of 0.13 nM.

The sensor was employed in spiked milk samples to evaluate the recovery. It ranged from 92.8% to 107.7% and can be considered satisfactory.

An aptamer cocktail was immobilized on the surface of gold screen-printed electrodes for developing an electrochemical aptasensor to detect tetracycline (TC) in honey [178], as shown in Figure 11. The aptamer cocktail was composed of a comparatively short aptamer (Apt40) and a comparatively long aptamer (Apt76), with a different base composition, different chain lengths, and differently folded binding sites.

The aptasensor provided a detection limit of 0.0073 ng mL^−1^ and a linear concentration range from 0.01 to 1000 ng mL^−1^. When detecting TC in spiked honey samples, the aptasensor showed high specificity and good recovery rates of 96.45–114.6%. This aptasensor can represent a model for developing aptasensors towards other targets.

Two typical kanamycin (KAN) electrochemical aptasensors employing different signal transduction mechanisms were designed and assembled with a similar structure [179]. One sensor (sensor-1) was based on the so-called classical probe conformation changing mode (PCCM) with a methylene blue (MB) label used as an electrochemical tag. The other sensor (sensor-2) used the target-induced signal probe shifting (TISPS) method with a free MB label in the solution. The difference in signal transduction mechanisms resulted in differences in electrochemical behavior and sensing performance. The results show that both sensor types exhibit different electrochemical behavior in square wave voltammetry, cyclic voltammetry, and sensitivity, with the detection limits of 3.0 nM for sensor-1 and 60.0 pM for sensor-2 in the buffer. When validated to detect tap water and milk samples, both sensing methods showed good performances with the detection limits of <260 nM and measurement times of <40 min. In addition, accuracy was good with mean recoveries of 72.3–92.6%.

Compared with PCCM, TISPS is more conducive with low background signals, improving the sensitivity, but has a little bit slower response and weaker anti-fouling ability in a complex matrix. Both sensors present their respective advantages, justifying their development to detect KAN.

A novel potentiometric aptasensor array based on a 4-channel screen-printed carbon electrode was developed with a dual-internal calibration system for the simultaneous detection of streptomycin (MLR 500 mg/kg in milk [170]) and kanamycin [180]. Two channels were used as working channels for assembling the aptamers of the two targets, and the other two channels were calibration channels.

Under optimal conditions, this aptasensor array showed a high sensitivity to detect streptomycin and kanamycin with the detection limits of 9.66 pM and 5.24 pM, respectively, and corresponding linear response ranges of 10 pM–10 μM and 10 pM–1 μM, respectively.

Moreover, it presented good specificity without interference between the two targets or with other antibiotics and also exhibited good repeatability. This aptasensor array was further applied to the simultaneous detection of streptomycin and kanamycin in real milk samples, and the results were validated by liquid chromatography-mass spectrometry (LC–MS).

### 3.5. Endocrine-Disrupting Chemicals

Endocrine-disrupting chemicals (EDCs) are environmental contaminants/pollutants and are also known as hormone-disrupting compounds [181]. The WHO is particularly sensitive to the problem of the presence and determination of endocrine disrupters [182].

Furthermore, EDCs represent a broad class of molecules, such as pesticides (see, for example, trifluralin [159]) and industrial chemicals, plastics and plasticizers, fuels, and many other chemicals present in the environment.

Herein, we focused our attention on bisphenol A (BPA) and on the estrogens and Table 5 summarizes the analytical characteristics of the electrochemical biosensors for BPA and estrogens reported in the review.

#### 3.5.1. Bisphenol A

Bisphenol A (BPA) is a synthetic chemical, classified as a non-biodegradable compound with high chemical resistance and widely used as a monomer in the synthesis of epoxy resins and polycarbonate.

Due to their properties, polycarbonates have different applications, such as in the fabrication of water bottles, infant feeding bottles, toys, utensils, thermal paper, and medical equipment. For similar reasons, epoxy resins are widely used as protective coatings for food and beverage containers, paints, adhesives, and electronic laminates. In both cases, BPA can contaminate food commodities and water.

Being an endocrine disruptor, BPA can cause serious adverse effects on human health even at very low concentrations [181,182]. BPA has a similar structure to that of estradiol and diethylstilbestrol and thus can stimulate a cellular response, binding with the estrogen receptors.

Because of its serious adverse effects, it is required to develop a reliable, remarkable selective and sensitive analytical method for BPA identification.

Recently, many efforts have been made to develop rapid, simple, sensitive, and field-portable alternatives for BPA detection, considering that the conventional methods require complex pretreatments of the sample, a long time for the analysis and skilled personnel.

Verdian presented [183] a comprehensive overview of recently developed aptamer-based biosensors to detect BPA. In addition, trends in developing colorimetric, fluorescence and electrochemical aptasensors for the monitoring of BPA are shown so that they can give a new idea for designing commercial kits.

A label-free impedimetric aptasensor to detect BPA was developed using a BPA-specific aptamer as a probe molecule [184]. The developed biosensor can detect BPA level in 20 s and exhibits a linear range from 1 fM to 10 pM, with a limit of detection of 152.93 aM. This biosensor was applied to test BPA in spiked canned food samples with good recovery results.

Li [185] developed an electrochemical impedance aptasensor based on an Au nanoparticle (Au-NPs)-coated boron-doped diamond (BDD) electrode modified with aptamers and 6-mercapto-1-hexanol (MCH) to detect BPA. It showed good linearity from 1.0 × 10^−14^ to 1.0 × 10^−9^ mol L^−1^. The detection limit of 7.2 × 10^−15^ mol L^−1^ was achieved, which can be attributed to the synergistic effect of combining BDD with Au-NPs, aptamers, and MCH. The results of BPA analysis in buffer and in milk indicated good sensitivity, specificity, stability, and repeatability of the aptasensor.

Another label-free electrochemical aptasensor was realized [186] to determine bisphenol A, based on functionalized multiwall carbon nanotubes/gold nanoparticles (f-MWCNTs/AuNPs) nanocomposite film modified gold electrode. Under the optimized experimental conditions, linear concentrations range from 0.1 to 10 nM with a detection limit of 0.05 nM. The effect of interfering species was investigated, and the proposed aptasensor resulted in selective BPA. In addition, the reproducibility and stability of the sensor were satisfactory. Finally, the developed aptasensor was successfully applied to real spiked samples, such as mineral water, orange juice, and milk, with results acceptable in terms of recovery

Zhang [187] prepared an electrochemical bisphenol A sensor using a hierarchical Ce-metal–organic framework (Ce-MOF) modified with cetyltrimethylammonium bromide (CTAB) as a sensing platform. Ce-MOF was synthesized and modified with a cationic surfactant (CTAB) via electrostatic interaction.

Metal–organic frameworks (MOFs), composed of metal ions and organic ligands connected each other through strong coordination bonds, have been widely applied in gas storage and separation due to their unique physical and chemical properties, such as tunable structure, ultra-high porosity, and high thermal and chemical stability [12]. In this work, cetyltrimethylammonium bromide (CTAB), a quaternary ammonium compound, and a cationic surfactant have been assembled onto hierarchical Ce-MOF to obtain a functional composite (CTAB/Ce-MOF) for preparing an electrochemical BPA sensor. The CTAB/Ce-MOF was prepared by modifying the CTAB monolayer on the surface of the Ce-MOF via the electrostatic interaction supported by the ultrasonication.

A CTAB/Ce-MOF composite suspension was dropped onto a glassy carbon electrode (GCE), to obtaining a final CTAB/Ce-MOF/GCE sensor. A linear concentration range from 0.005 to 50 μmol L^−1^ and a low detection limit of 2.0 nmol L^−1^. The proposed sensor showed good reproducibility, stability, and anti-interference behavior and was applied for BPA detection to spiked milk samples, with acceptable recovery results ranging from 96.2 to 104.6%.

An interesting electrochemical sensor for BPA based on the AuPd nanoparticles incorporated in carboxylic multi-walled carbon nanotubes (MWCNT) was designed and assembled by Liu [188], where MWCNT improved electron transfer and poly-(diallyldimethylammonium chloride) (PDDA) acted as a dispersing agent for MWCNT and for further increasing the metal NPs loading. A linear concentration range of 0.18–18 μM, and the detection limit of 60 nM was determined. The sensor showed good sensitivity, stability, repeatability and can detect BPA in spiked milk and water samples with good performance in terms of recovery.

A sensitive electrochemical aptasensor was developed [189] to detect BPA based on MWCNT/SiO_2_@Au nanocomposite. The detection strategy is based on [Fe (CN)_6_] ^3−/4−^ as a label-free redox probe. In the absence of BPA, the aptamers remain to unfold. After the BPA addition, strong interactions between the analyte and the aptamer are evidenced, and an electrochemical signal decrease occurs. The proposed electrochemical aptasensor was selective with a linear concentration range from 0.1 to 100 nM and a limit of detection of 10 pM. This aptasensor was successfully applied to detect BPA to spiked water, orange juice and milk samples obtaining acceptable recovery results ranging from 96 to 104%.

Graphene nanoplatelets (GNPs), multiwalled carbon nanotube (MWCNTs) and chitosan (CS) were self-assembled by a one-step hydrothermal reaction and a novel MWCNTs–CS enfolded GNPs (GNPs–MWCNTs–CS) composite was synthesized and used to modify a glassy carbon electrode (GCE).

The GNPs–MWCNTs–CS/GCE was employed as a sensing platform to determine BPA by differential pulse voltammetry (DPV) [190]. Under the optimum conditions, a linear current concentration range from 0.1 to 100 μM with a detection limit of 0.05 nM is observed.

The proposed sensor showed good selectivity, repeatability, and reproducibility, and it was applied to different spiked milk samples with interesting results in terms of recovery.

As a last-but-not-least example, a three-dimensional hierarchical cylinder-like nickel nanoparticle/nitrogen-doped carbon nanosheet/chitosan nanocomposite (NiNP/NCN/CS) was used for modifying a glassy carbon electrode (GCE) to assemble a sensing platform for BPA determination [191].

Two linear concentration ranges were observed, the first from 0.1 to 2.5 μM and the second from 2.5 to 15.0 μM. The detection limit for BPA detection is estimated to be 45 nM. Finally, the BPA sensor showed good selectivity and stability, and it was employed to detect BPA in spiked milk samples, with recoveries ranging from 96 to 105%.

#### 3.5.2. Estrogens

Among the endocrine-disrupting chemicals, environmental estrogen is a type of endocrine disruptor, able to interfere with the hormone metabolism in human organisms, thereby affecting physiological functions, such as growth, development, and reproduction. Typically, environmental estrogens are divided into naturally produced, such as 17b-estradiol (E2), estrone (E1), and estriol (E3), and synthetic forms, such as bisphenol A (BPA) [172,173], 17a-ethinylestradiol (EE2), diethylstilbestrol (DES), and others

Natural estrogens are synthesized in the human and animal organism, and synthetic estrogens are generally employed as active agents in contraception or hormone therapy [192]. These estrogens can penetrate the waterways after excretion from humans and animals, adding to the natural estrogens and veterinary drugs excreted by livestock in many rural areas. In this context, it is inevitable that estrogens can access the food chain and affect public health. Due to their lipophilicity, estrogens can accumulate in the adipose tissues [193]. The exogenous estrogens are very slowly eliminated, so interfering with the function and metabolism of the endocrine system. Therefore, it is of great importance to monitor exogenous estrogen contamination in water and food.

Very recently, interesting reviews focused on methods concerning the determination of estrogens in food matrices have been published.

In particular, Gunatilake [194] revised the novel methodological developments to determine five steroidal estrogens, estriol, 17a-estradiol, 17b-estradiol, estrone, and ethinyl estradiol in food matrices including dairy products, fish, meat. In addition, significant attention has been given to methods and analytical approaches, which allow to directly determine the contaminant, optimizing analysis time and protocols.

Bala [195] reported an overview concerning the recent advances in electrochemical sensors based on electrochemical impedance spectroscopy to detect endocrine disruptors, including synthetic estrogens. In this review, the fact that EIS -based sensors can be easily implemented in fully automated devices by integrating electrodes in microfluidic chips has been emphasized.

Finally, Sun [196] extensively described the recent developments in biosensors to detect estrogens in the environment and food, including molecule-based biosensors, cell-based biosensors, and model organism-based biosensors.

In particular, works published in 2017–2019 and focused on methods to detect estrogens and using nanomaterials for biosensors development have been considered, evidencing the advantages and limitations of the different kinds of biosensors.

Herein, we reported some newly released examples of electrochemical (bio)sensors for estrogen determination.

Oliveira [197] and coworkers described a promising electrochemical sensor to monitor the synthetic estrogen E,E-dienestrol (E,E-DNL) in fish tissue, using a cathodically pretreated boron-doped diamond (Cpt-BDD) electrode combined with the quick, easy, cheap, effective, rugged and safe (QuEChERS) extraction method.

A linear concentration range from 2.30 × 10^−7^ to 9.69 × 10^−6^ mol L^−1^ of E,E-DNL, with a detection limit of 5.43 × 10^−8^ mol L^−1^, good repeatability and reproducibility test was evidenced. The procedure was successfully applied to quantify E,E-DNL in QuEChERS extracts from Nile tilapia (*Oreochromis niloticus*) liver tissue with a recovery ranging from 92.3 to 98.8%.

**Table 5 molecules-26-02940-t005:** An overview of recent electrochemical biosensors for BPA and estrogens determination.

Electrode	(Bio)Sensor Format	Electrochemical Technique	Sample/Analyte	L.R.	LOD	References
Interdigitated electrode (IDE)	Label-free impedimetric aptasensor printed circuit board (PCB) technique	EIS	BPA/canned food	1 fM–10 pM	152.93 aM	[184]
BDDE	Impedimetric aptasensor based on Au nanoparticles (Au-NPs) coated boron-doped diamond (BDD) modified with aptamers, and 6-mercapto-1-hexanol (MCH)	EIS	BPA/milk	1 × 10^−^^14^–1 × 10^−^^9^ M	7.2 × 10^−15^ M	[185]
AuE	Label-free electrochemical aptasensor based on functionalized multiwall carbon nanotubes/gold nanoparticles (f-MWCNTs/AuNPs) nanocomposite film modified gold electrode	SWV	BPA/mineral water, orange juice, milk	0.1–10 nM	0.05 nM	[186]
GCE	Electrochemical sensor using hierarchical Ce-metal–organic framework (Ce-MOF) modified with cetyltrimethylammonium bromide (CTAB) as a sensing platform	DPV	BPA/milk	0.005–50 mM	2 nM	[187]
GCE	Electrochemical sensor based on the AuPd nanoparticles incorporated carboxylic multi-walled carbon nanotubes (MWCNT)	DPV	BPA/milk	0.18–18 μM	60 nM	[188]
GCE	Electrochemical aptasensor based on MWCNT/SiO2@Au nanocomposite	SWV	BPA/water, orange juice, milk	0.1–100 mM	10 pM	[189]
GCE	Electrochemical sensor using as sensing platform multi-walled carbon nanotubes and chitosan (MWCNTs–CH) self-assembled on graphene nanoplatelets GNPs (GNPs–MWCNTs–CH)	DPV	BPA/milk	0.1–100 μM	0.05	[190]
GCE	Electrochemical sensor based on three-dimensional hierarchical cylinder-like nickel nanoparticle/nitrogen-doped carbon nanosheet/chitosan nanocomposite (NiNP/NCN/CHI)	DPV	BPA/milk	0.1–2.5 mM and 2.5–15.0 mM	45 nM	[191]
BDDE	Electrochemical sensor, using a cathodically pretreated boron-doped diamond (Cpt-BDD) electrode combined with QuEChERS extraction method	SWV	E,E-dienestrol/fish tissue	2.30 × 10^−7^–9.69 × 10^−6^ M	5.43 × 10^−8^ M	[197]
SPCE	Impedimetric aptasensor based on carbon nanodots modified SPC electrode	EIS	17b-estradiol/water	1.0 × 10^−7^–1.0 × 10^−12^ M,	0.5 × 10^−12^ M.	[198]
AuE	Electrochemical biosensor based on graphene quantum dots (GQDs)/conducting polymer and laccase modified gold electrodes	CV	17b-estradiol/no real samples	0.1–120 × 10^−6^ M	1 mM	[199]

Abbreviations: AuE: gold electrode; AuNPs: gold nanoparticles; AuSPE: gold screen-printed electrode; BDDE: boron-doped diamond electrode; BPA: bisphenol A; CA: chronoamperometry; CPE: carbon paste electrode; CH: chitosan; CF: carbon felt; CNF: carbon nanofibers; CV: cyclic voltammetry; DEP: disposable electrical printed microarray electrode; DPV: differential pulse voltammetry; EIS: electrochemical impedance spectroscopy; ECL: electrochemiluminescence; GQDS: graphene quantum dots; GCE: glassy carbon electrode; GO: graphene oxide; IDE: interdigitated electrode; ITO: indium–tin-oxide electrode; MBs: magnetic beads; MIPs: molecularly imprinted polymers: MWCNTs: multi-walled carbon nanotubes; NPG: nanoporous gold; QDs: quantum dots; PCNB: Printex carbon nanoballs; PGE: pencil graphite electrode; QuEChERS: quick, easy, cheap, effective, rugged and safe; SPCE: screen-printed carbon electrode; SWV: square-wave voltammetry.

Haniphah [198] designed and assembled a simple and sensitive impedimetric aptasensor based on conductive carbon nanodots (CDs) to detect 17b-estradiol (E2). Carbon nanodots were electrodeposited on a screen-printed electrode (SPE), acting as a platform for immobilizing 76-mer aptamer probe. Figure 12 shows the process step-flow for the fabrication and assembling of impedimetric aptasensor to determine 17b-estradiol.

The impedimetric aptasensor exhibited a linear concentration range from 1.0 × 10^−7^ to 1.0 × 10 ^−12^ M, with a detection limit of 0.5 × 10^−12^ M. The developed biosensor showed high selectivity toward E2 in the presence of progesterone (PRG), estriol (E3) and bisphenol A (BPA), respectively.

Moreover, the average recovery rate for spiked river water samples ranged from 98.2% to 103.8%, evidencing the aptasensor possible application for E2 detection in water samples.

Finally, two biosensors based on graphene quantum dots (GQDs)/laccase gold (Au) electrodes were developed by Cabaj and coworkers [199]. The process of hormone determination was based on the redox reaction catalyzed by the laccase enzyme.

Under optimized conditions, the biosensor showed a linear range from 0.1–120 × 10^−6^ M) with a detection limit of about 1 μM. Moreover, the method was successfully applied for hormone determination in the presence of interfering compounds, such as ascorbic acid, L-cysteine, uric acid. As a final comment, an investigation of the biosensor applicability to real samples, e.g., in water, should be important.

### 3.6. Allergens

Anomalous reactions due to food ingestion are defined as “adverse reactions to food”. They are classified by the European Academy of Allergology, and Clinical Immunology, based on the response mechanism, as toxic and nontoxic reactions [200]. Toxic reactions are connected with a food’s primary harmful effect after ingestion. Nontoxic reactions depend on individual sensitivity, are not commonly dose-dependent, and are classified as immunological (food allergy) and non-immunological (food intolerance) [201,202]. Food allergy is an adverse immune-mediated response occurring after an ingestion/exposure to a given food, component, or ingredient.

Food allergens represent a major food safety concern in industrialized countries.

The European Union has established labeling rules for 14 allergenic food ingredients, i.e., eggs, milk, peanuts, nuts, gluten-containing cereals, lupin, soybeans, celery, mustard, sesame seeds, fish, crustaceans, mollusks, and sulfites: therefore, it is mandatory to label them on their food derivatives [203] Although food labeling is required for providing consumers with composition information, accidental ingestion/exposure to some allergen can occur. This exposure can be due to undeclared allergens through adulteration, cross-contamination, or even fraud.

Considering the scenario described above, it is clear that precise, cost-effective and fast analytical methods are required for reliable screening of specific allergens in food commodities and electrochemical biosensors seem to meet all these requirements, including on-site analysis and involving unskilled personnel.

Several reviews reported the (bio)sensing approaches to determine food allergens.

In particular, Marty and coworkers [204] highlighted the success of applying electrochemical affinity biosensors based on disposable screen-printed electrodes to detect allergens, also reporting some interesting examples for specific allergens.

Pingarron and his group [205] presented the most significant trends and developments in electrochemical affinity biosensing in this field over the past two years, as well as the challenges and future prospects for this technology.

Conte-Junior [206], in his review, underlined that integrating biosensors and nanoparticles is very promising for the accurate and reliable analysis of allergenic proteins in the food samples.

Finally, Maquieira [207] reviewed recent approaches, including the electrochemical biosensors, existing kits for foodborne allergen detection and cutting-edge applications by focusing on the sensitivity, selectivity, and applicability of current methods in food samples.

Herein, we reported significant examples of electrochemical (bio)sensors to detect allergens, and Table 6 summarizes the analytical characteristics of the electrochemical biosensors reported in the review.

#### 3.6.1. Gliadin

The starting examples are focused on the design and assembling of biosensors for detecting the protein gliadin, responsible for a serious autoimmune disorder causing chronic diarrhea, fatigue, weight loss and anemia in celiac people.

The first example is a biosensor where natural polymer zein, coupled with nanomaterials, such as carbon nanotubes, acts as a natural platform for anchoring the capture antibody onto the glassy carbon electrode (GCE) [208]. GCE was functionalized through a layer-by-layer deposition of zein and carbon nanotubes (Z-CNT) nanocomposite, where Z-CNT behaves as a natural linker molecule with several functional groups for immobilizing capture antibody and target, guaranteeing good sensor performances.

The Z-CNT biosensor showed a detection limit of 0.5 ppm. Linear concentration range from 5 to 100 ppm, good selectivity for gliadin towards other food toxins and good stability. In addition, it was also applied to wheat flour samples, the content of gliadin was examined after its extraction from the flour samples, and the extracts were analyzed with acceptable results in terms of accuracy and specificity.

Singh [209] proposed a microfluidic electrochemical aptasensing device to detect gliadin. The sensor assembling involves the combining use of a 2D nanocomposite involving molybdenum disulfide (MoS_2_)/graphene and gold nanoparticles. Aptamers, specific for gliadin, were used as biomarkers. A polydimethylsiloxane (PDMS)-based flexible microfluidic device integrated the sensor. The aptasensor showed a limit of detection was 7 pM, and a good linear range was observed from 4 to 250 nM. Samples of rice flour, naturally gluten-free, spiked with wheat flour, were tested, and good recovery was observed between 98 and 102%.

#### 3.6.2. Milk Allergens

Among the food allergies, cow milk allergy is one of the most common forms of childhood allergy, and unfortunately, this kind of allergy can persist for life.

Marrazza and her group have recently published an interesting review and some papers concerning the detection of milk allergens using electrochemical biosensors.

The review [210] focused on particular research advances in biosensors (specifically immunosensors and aptasensors) to detect milk allergens. Different allergic proteins of cow milk are described together with the analytical standard methods for their detection. Additionally, the commercial status of biosensors is also discussed compared to conventional techniques like enzyme-linked immunosorbent assay (ELISA).

The same group developed [211] a disposable electrochemical platform based on poly(aniline-co-anthranilic acid) (PANI/PAA) copolymer coupled with an aptamer to detect *β*-lactoglobulin, the main cause of the milk infant allergy. After optimizing the experimental parameters, a dose–response curve was obtained between 0.01 and 1.0 μg mL^−1^
*β*-lactoglobulin concentration range with a limit of detection of 0.053 μg L^−1^. Milk samples spiked with *β*-lactoglobulin were analyzed with a recovery range between 80 and 95%.

**Table 6 molecules-26-02940-t006:** An overview of recent electrochemical biosensors for allergens determination.

Electrode	(Bio)Sensor Format	Electrochemical Technique	Analyte/Sample	L.R.	LOD	References
GCE	Electrochemical immunosensor based on zein polymer coupled with carbon nanotubes as a sensing platform to immobilize capture antibody	SWV	Gliadin/wheat flour	0.5–100 ppm	0.5 ppm	[208]
SPCE	Microfluidic electrochemical aptasensing system based on a combination of 2D nanomaterial molybdenum disulfide (MoS_2_) and graphene with the addition of gold nanoparticles	DPV	Gliadin/wheat flour	4–250 nM	7 pM	[209]
GSPE	Electrochemical aptasensor based on poly(aniline-*co*-anthranilic acid) (PANI/PAA) composite polymer coupled with a specific aptamer	DPV	*β*-lactoglobulin/milk	0.01–1.0 μg mL^−^^1^	0.053 μg mL^−1^	[211]
GSPE	Electrochemical aptasensor based on poly-L-lysine modified graphite electrodes	DPV	*β*-lactoglobulin/milk, yogurt	0.1–10 ng mL^−1^	0.09 ng mL^−^^1^.	[212]
SPAuE	Electrochemical label-free immunosensor using polypyrrole (PPY) electropolymerized as immobilizing platform for the capture antibody	DPV	*a*-lactoglobulin/meal	355–2840 pg mL^−1^	0.192 fg mL^−1^	[213]
ITO	Electrochemical aptasensor based on a highly selective DNA aptamer and flower-like Au@BiVO_4_ microspheres	Amperometry	*β*-lactoglobulin/infant food formula	0.01–1000 ng mL^−1^	0.007 ng mL^−1^	[214]
GSPE	Electrochemical immunosensor based on gold-nanocluster-modified graphene screen-printed electrodes	DPV	*β*-lactoglobulin/milk	0.01–100 ng mL^−1^	0.08 ng mL^−1^	[215]
PGE	Electrochemical sensor based on graphene-oxide modified pencil graphite electrode	CV	*β*-lactoglobulin/milk	530–11.160 Mg L^−1^	270 mg L^−1^	[216]
SPCE	Disposable amperometric magnetoimmunosensor using a sandwich configuration involving selective capture and detector antibodies and carboxylic acid-modified magnetic beads (HOOC-MBs)	Amperometry	Ara h 2/flour	87–10.000 pg mL^−^^1^	26 pg mL^−1^	[217]
GCE	DNA biosensor based on gold–palladium nanowaxberries (AuPd NWs)/dodecylamine functionalized graphene quantum dots (D-GQDs)-graphene micro-aerogel (GMA) composite	DPV	Ara h 1/peanut milk	1.0 × 10^−22^–1.0 × 10^−17^ M	4.7 × 10^−23^ M	[218]
SPCE	Paper-based capacitance mast cell sensor based on 3D paper chip printed with carbon electrodes as a noncontact capacitance sensing platform, using a polyvinyl alcohol (PVA)-gelatin methacryloyl (GelMA)-nano-hydroxyapatite (nHAP) composite hydrogel (PGHAP gel) to improve the conductivity and biocompatibility of the cellulose paper	Capacitance measurement	Ara h 2/raw and fried peanut	0.1–100 ng mL^−1^	0.028 ng mL^−1^	[219]
Magnetic glassy carbon electrode (MGCE)	Cell sensor, based on fluorescent magnetic beads	EIS	Tropomyosin and parvalbumin/crucian carp and brown shrimp	-	Tropomyosin 0.03 μg mL^−1^ Parvalbumin 0.16 ng mL^−1^	[220]
SPCE	Label-free electrochemical immunosensor assembled by electrochemically reducing 4-carboxyphenyl diazonium salt, which was electrochemically generated in situ, to a stable 4-carboxyphenyl layer on carbon nanofiber-modified screen-printed electrode	DPV	Porcine serum albumin/pork fresh meat	0.5–500 pg mL^−1^	0.5 pg mL^−1^	[221]
CE	Electrochemical sensor using molecularly imprinted polymers (MIPs) for detecting genistein, an allergenic soy marker	DPV	Genistein/no real samples	100 ppb-10 ppm	100 ppb	[222]

Abbreviations: AuE: gold electrode; AuNPs: gold nanoparticles; AuSPE: gold screen-printed electrode; CA: chronoamperometry; CE: Carbon electrode; CPE: carbon paste electrode; CH: chitosan; CF: carbon felt; CNF: carbon nanofibers; CV: cyclic voltammetry; DEP: Disposable electrical printed microarray electrode; DPV: differential pulse voltammetry; EIS: electrochemical impedance spectroscopy; ECL: electrochemiluminescence; GQDS: graphene quantum dots; GCE: glassy carbon electrode GSPE: graphene screen-printed electrode; GSPE: graphite screen-printed electrode; GO: graphene oxide; IDE: interdigitated electrode; ITO: indium–tin-oxide electrode; MGCE: magnetic glassy carbon electrode; MBs: magnetic beads; MIPs: molecularly imprinted polymers: MWCNTs: multi-walled carbon nanotubes; NPG: nanoporous gold; QDs: quantum dots; PANI: polyaniline; PAA: poly(anthranilic acid); PGE: pencil graphite electrode; SPCE: screen-printed carbon electrode.

Another sensing methodology always to determine *β*-lactoglobulin in food samples was designed by the Marrazza group [212] using a folding-based electrochemical aptasensor based on poly-L-lysine modified graphite screen-printed electrodes (GSPEs) and an anti-*β*-lactoglobulin aptamer tagged with methylene blue (MB). This aptamer changes its conformation when the sample contains β-LG, and this induces changes in the distance between MB and the electrode surface and consequently in the electron-transfer rate, as illustrated in Figure 13.

The response of this biosensor was linear for concentrations of β-LG within the range 0.1–10 ng mL^−1^, with a limit of detection of 0.09 ng mL^−1^.

The aptasensor performance was evaluated on spiked food samples: biscuits and soya yogurt, with a recovery range from 95 to 117%.

Carrara [213] presented a voltammetric label-free aptasensor to detect alpha-lactalbumin (α-LB) in meal samples. The sensing strategy is based on capturing of α-LB via entrapped α-LB antibody (α-LB-Ab) through electropolymerization of polypyrrole (PPy) and then measuring the conductivity decrease by differential pulse voltammetry (DPV). A limit of detection of 0.19 fg mL^−1^ was obtained with a linear concentration range from 355 to 2840 pg mL^−1^. The aptasensor was applied to detect α-LB in real spiked samples of different kinds of milk (UHT whole milk, low-fat milk, dry milk, and almond milk) with a recovery ranging between 93 and 97%.

An electrochemical biosensor to detect *β*-lactoglobulin was developed by Huang [214]. A DNA aptamer was used instead of an expensive antibody as the recognition element for *β*-lactoglobulin. The flower-like BiVO_4_ microspheres were employed because they mimic the peroxidase catalytic activity and can amplify the electrochemical signal. This electrochemical biosensor exhibited a detection range from 0.01 to 1000 ng mL^−1^, with a limit of detection of 0.007 ng mL^−1^. The biosensor was applied to determine *β*-lactoglobulin in spiked infant food formula with a recovery ranging from 92 to 103%.

An electrochemical immunosensor based on modified screen-printed electrodes (SPEs) was designed to detect *β*-lactoglobulin (β-Lg) [215]. The surface modification of SPEs was accomplished by a simple drip coating using polyethyleneimine (PEI), reduced graphene oxide (rGO), and gold nanoclusters (AuNCs), and the obtained SPEs showed a good electrical conductivity. An anti-β-Lg antibody was then immobilized on the nanocomposite, inducing a reduction in SPEs conductivity due to the reaction between antigen and antibody. The sensor showed a limit of detection (LOD) of 0.08 ng mL^−1^ and a detection range from 0.01 to 100 ng/mL^−1^ for β-Lg. Furthermore, milk samples from four milk brands were analyzed, and the results agreed with those from ELISA.

Finally, Abaci [216] developed a graphene-oxide-modified pencil graphite electrode to determine *β*-lactoglobulin. A linear concentration range of 0.53–11.16 mg mL^−1^ with a detection limit of 0.27 mg mL^−1^. The sensor was applied to spiked milk samples, obtaining recoveries between 118.30 and 90.00%.

#### 3.6.3. Peanut Allergens

Peanut allergy is a frequent cause of serious anaphylactic reactions and severe diseases among food allergies.

The detection of peanut allergens in food products is sometimes challenging since they are often present unintentionally and in trace amounts or can be masked by other compounds of the food matrix [217]. Different methods are available for the peanut allergens detection, among them, those based on immunoassay (ELISA) are well-known and appreciated for their specificity and sensitivity, but they showed some drawbacks, i.e., cross-reaction, long analysis time, high cost of ELISA kits, and large numbers of sample replications. Biosensors have become attractive compared with the conventional approaches, providing real-time, possibly on-site, cost-effective, and high-sensitivity analysis.

Pingarron group developed a disposable amperometric magnetoimmunosensor for the rapid determination of Arah 2 protein, one of the major peanut allergens [217]. The approach used a sandwich configuration involving capture and detector antibodies and carboxylic acid-modified magnetic beads (HOOC-MBs). Detector antibodies are marked with HRP-conjugated secondary antibodies, and the MBs bearing the immunoconjugates are magnetically captured on the surface of a disposable screen-printed carbon electrode (SPCE). The immunosensor showed a linear concentration range from 87 to 10,000 pg mL^−1^—with a detection limit of 26 pg mL^−1^ and good selectivity towards possible interferent other proteins. The sensing platform was applied to detect Ara h 2 in different food extracts. After an appropriate sample dilution, no matrix effects were evidenced. The developed methodology determined trace amounts of the peanut allergen (0.0005% or 5.0 mg kg^−1^) in wheat flour spiked samples, and the obtained results agreed with those of the ELISA kit.

Li [218] reported the synthesis of gold–palladium nanowaxberries (AuPd NWs)/dodecylamine functionalized graphene quantum dots (D-GQDs)-graphene micro-aerogel (GMA). The AuPd NWs/D-GQDs-GMA hybrid composite shows a particular three-dimensional architecture, improving the amplification of the detection signal significantly.

A DNA biosensor for peanut allergen Ara h 1, based on this hybrid nanocomposite, was assembled and exhibited a linearity range from 1.0 × 10^−22^ to 1.0 × 10^−17^ M with the detection limit of 4.7 × 10^−23^ M. This sensing platform was applied to the determination of peanut allergen Ara h 1 in spiked peanut milk with a corresponding recovery of 96.7%.

An innovative paper-based capacitance mast cell sensor was designed and developed by Wang [219] for real-time monitoring of the peanut allergen Ara h 2.

A noncontact capacitance sensing platform was fabricated, employing a 3D paper chip printed with carbon electrodes. To improve the conductivity and biocompatibility of the paper chip, polyvinyl alcohol (PVA)-gelatin methacryloyl (GelMA)-nano-hydroxyapatite (nHAP) composite hydrogel (PGHAP gel) was introduced. When rat basophilic leukemia mast cells (RBL-2H3) are immobilized and cultured on the 3D paper modified chip, signals of Ara h 2 were specifically monitored in real time by capacitance change measurement.

A dose-dependent response for the allergen determination was obtained in the concentration range from 0.1 to 100 ng mL^−1^. Finally, the capacitance cell sensor performance was assessed using raw peanuts and fried peanut extracts analysis. The obtained results agreed with those obtained with the conventional methods.

Previously another example of a mast cell-based electrochemical sensor to detect different allergens in foodstuffs was developed, using the same rat basophilic leukemia cells and fluorescent magnetic beads [220].

Results showed that the exposure of model antigen–dinitrophenol–bovine serum albumin (DNP–BSA) to anti-DNP IgE-sensitized mast cells induced an electrochemical impedance dose-dependent signal. The detection limit was identified at 3.3 × 10^−4^ ng mL^−1^. To demonstrate the possible application of this biosensor to real food commodities, it was employed to quantify both shrimp allergen tropomyosin (Pena1) and fish allergen parvalbumin (PV). Results show accuracy for these targets, with a limit of 0.03 μg/mL (shrimp Pena1) and 0.16 ng/mL (fish PV), respectively.

A label-free electrochemical immunosensor for sensitive detection of porcine serum albumin (PSA) was developed by Ahmed [221], using a stable 4-carboxyphenyl layer deposited on a carbon nanofiber-modified-screen-printed electrode. Antibodies were covalently attached to the electrode. The immunosensor exhibited a linear range from 0.5 to 500 pg mL^−1^ with the detection limit of 0.5 pg mL^−1^ in buffer solution.

Cross-reactivity studies have shown good specificity with the satisfactory recovery of PSA in fresh meat samples without sample dilution.

An electrochemical device using poly(o-phenylenediamine) as a molecularly imprinted polymer (MIPs) [222] could detect allergenic soy markers, such as genistein, with a detection limit of 100 ppb, concentration known to produce an adverse effect in patients. On the other hand, the sensor performance was only qualitatively validated with commercially available soy allergen detection lateral flow devices (LFDs). The MIP-sensors correctly identified the presence or absence of the genistein, with 100% accuracy in all food samples. It seems that the result of this sensor application can address a peculiar analytical challenge to achieving fast, cost-effective, and qualitative methods for direct detection of allergen tracers in food analysis.

## 4. Conclusions, Challenges, and Future Perspectives

The development of highly sensitive, reliable, robust, portable, and cost-effective sensing approaches has become fundamental to guarantee food safety, addressing the critical issue of infection/contamination of food commodities due to several causes, such as bacteria, contaminants, allergens, drugs, etc.

Considering the drawbacks of the conventional analytical approaches, such as complex analytical protocols, long duration of the analytical procedure, costly operation, and skilled personnel, it is quite clear that the biosensing approach is very attractive for many reasons: easy to handle, relatively low cost, good sensitivity, and easy miniaturization.

In this area, electrochemical biosensors are emerging sensing tools, especially if nanomaterials can improve the analytical performances. However, several issues and challenges should be faced.

For example, some described analytical protocols involve using sensitive reagents and multiple-step procedures, which increase measurement time and cost, making their introduction into the food safety and regulatory field very complicated.

Moreover, most of the described assays have addressed target analyte quantification just in aqueous solutions or synthetic samples (prepared by adding the contaminant in an intermediate step or even at the end of sample preparation), and only a few analyze real samples.

There are two relevant issues associated with real sample analysis: possible electrochemical interferences and efficient extraction of target analytes from the complex food matrix. To avoid electrochemical interferences, surface chemistry and type of bioreceptor need to be carefully optimized, combined with sample pretreatment and cleanup.

Analyzing the data presented in this review, most electrochemical biosensors belong to the class of affinity biosensors. In particular, the number of aptasensors is particularly high. In few words, considering a schematic illustration of an aptasensor functionality, aptamers are confined at the electrode surface. An alteration in configuration of immobilized aptamers due to the reaction of aptamers with the target analyte induces changes in the recorded electrochemical signal in terms of potential, current, conductance, or impedance. This change in analytical signal could be used for sensing the target analyte. Such aptasensors provide high sensitivity, inexpensive, and unique specificity of aptamers with the target analyte.

Several examples of immunosensors have also been presented. In general, this approach involves the direct binding of an antibody to an analyte to form an immunocomplex at the electrode surface, and the changes in the surface potential and oxidation state of an electroactive species were recorded. Some authors stated that using an aptamer for realizing an aptasensor is less expensive than one of the corresponding antibodies (see, for instance, reference [214], justifying the preference for the aptasensing approach.

Very few examples involving the biocatalytic electrochemical sensor, i.e., the classical enzyme-based sensors. It is reasonable to assume that the lack of sufficient selectivity of enzymes for a particular analyte or the analyte not being commonly found in living systems is the principal reason for this behavior. Hence, affinity biosensors are considered a good option.

The bioreceptors, such as enzymes, antibodies, DNA, and aptamers, representing the specificity key element for the developed sensor towards the analyte, are a critical issue. The sensitivity of the bioreceptor is limited by the immobilization protocol of the biomolecule onto the electrode surface without affecting its biological activity. Such improvements can improve the stability and overall life of biosensors.

It is worthy to evidence that, in different examples of sensors present in this review, the bioreceptor is not present at all (for instance, see Section 3.3), but it was substituted by a nanomaterial and/or nanocomposite mimicking the bioreceptor action or activity. In this way, the issues linked to the immobilization protocols could be solved, but accurate studies and analyses of the toxicity and degradation of these nanomaterials are required.

In particular, it should be mentioned that the introduction and the wide use of nanomaterials must address these criticalities: (i) the sustainability of nanostructures in sensor applications, which have been insufficiently investigated, (ii) the sustainable fabrication of nanostructures, and (iii) the toxicity, which changes according to the physical properties of the material type.

As a general comment, nanomaterials, nanoparticles, nanocomposites, and nanostructures, used both as electrode modifiers and for electrochemical signal amplification, play an important role in developing electrochemical biosensors for food safety with improved performance in terms of stability and sensitivity.

Moreover, apart from the conventional classification of biosensors based on bioreceptors, nanomaterials such as nanotubes, quantum dots, etc., can be considered a new typology of bioreceptors. With the advancement of nanotechnology and nanoscience, different nanomaterials have been used as bioreceptors, opening a wider range of applications in biosensing technology. Nanomaterials can act as bioreceptors as well as transducers.

Presented in this review, several examples of applying screen-printed electrochemical sensors, many of them involving applying nanomaterials, demonstrate that it is a very important field of food analysis.

To summarize, the screen-printed technology applied to electrochemical biosensors in food quality control provides important features, such as miniaturization of the measuring setups, a low-cost of mass production, easy procedures of the use, and also the possibility of using such devices with small sample volumes. Further, a proper selection of nanomaterials employed for their construction can also improve response time and enhance selectivity and sensitivity.

On the other hand, analyzing the examples presented in this review, paper-based electrochemical sensors have not found wide application in the food analysis sector. In fact, only one is present [219], involving allergens detection. It must be emphasized that this kind of sensor has attracted extensive attention because of the advantages of sensitivity and selectivity. Moreover, these sensors can be miniaturized and easily fabricated on top of the paper. They represent a promising platform for lab-on-paper devices, where large-scale and complicated laboratory tests can be easily performed. In addition, they are inexpensive, portable, sensitive, selective, and on-site real-time detection devices and can be considered interesting in many application areas, such as food analysis and control. For improving the applicability of paper-based sensors, future efforts should be directed towards: (a) enhancement of fabrication and modification techniques for paper substrates and electrode materials; (b) further development of different detection platforms; (c) developing integrated and automated devices; (d) developing simpler and low-cost mass fabrication methods that for a possible introduction into the sensors market; and (e) the consideration of self-powered devices, which could extend the on-site applications.

Sample preparation and efficient extraction of the targets remain the steady steps, limiting the total analysis time and biosensor final performance.

Moreover, more rigorous validation studies are required, and the storage and operational stability of the electrochemical biosensor under real analysis conditions should be efficiently addressed and tested.

Additional work should also be performed to apply these biosensors to the analysis of food processed samples to evaluate the impact of food processing on their detection capability.

Another important issue of electrochemical biosensors is the capability to simultaneously measure and discriminate different analytes in a sample, and it should be attractive for commercial exploitation.

The implementation of multianalyte methodologies implies significant advantages over single analyte tests in terms of cost per assay, work loading, assay throughput and suitability. As a general strategy, electrochemical approaches for the multiple and simultaneous detection of pathogenic bacteria, toxins, pesticides, contaminants, and allergens involve sensing platforms and devices using principles and methodologies from immuno- and genosensors, as well as from other approaches introducing nanomaterials. Hybrid nanocomposites ad hoc synthesized coupled to computerized data analysis.

Different examples of multianalyte detection are introduced in this review, involving bacteria pathogens, pesticides, and antibiotics detection [114,139,162,166,173,181].

To assemble multiplex electrochemical biosensors, it is mandatory to provide some encoded probes, which can specifically capture different targets and then convert them to corresponding electrochemical signals simultaneously. Thus, specific target-recognition components, specific electrochemical markers coding for each analyte, and disguisable electroactive substances are needed. It should be noted that the examples are few, and several issues and aspects must be addressed, such as optimizing user interfaces and sample handling, using micro- and nano-fabrication techniques enabling the performance of multianalyte analysis with the same device, developing parallel computational methods to convert electronic responses for each analyte into concentration data, and integrating these multianalyte platforms into portable systems.

In addition, developing smart sensors is linked to developing portable devices. Improved portability may be achieved by integrating electrochemical biosensors with devices like smartphones and tablets, but very few examples are available [137].

Such integration of two distinct areas of research (sensors and ICT), addressing the day-to-day needs of people, could facilitate introducing the next generation of smart sensors into the food processing industries to increase the quality and safety of food and beverages.

As concluding remarks, although biosensors display clear advantages over traditional methods, a perfect biosensing technique does not yet exist, and there are many difficulties in its development to be overcome.

Many of the biosensors suggested in this review can be used in early disease monitoring and food controlling in laboratories due to their various applications with a low limit of detection values. Consequently, they present a great potential for commercialization. On the other hand, transforming biosensor technology into a market product from lab-scale research is still blocked by different shortcomings. Foremost, the demand for developing more sensitive biological sensing layers has urged researchers to design highly complex and sophisticated materials, which becomes an extremely expensive item.

The stability of the biological receptors immobilized in these complex structures is crucial in real sample analysis, representing an obstacle preventing biosensor commercialization. Research activities in artificial receptors, such as aptamers, have increased in the last few years, but more efforts must be made in this field. Nevertheless, it is almost certain that the future of electrochemical biosensors will involve partnership with information and communications technologies to assist food producers, retailers, authorities, and even consumers, in their decision-making process by equipping them with the necessary tools. The combination of different types of biosensors or hybrid biosensors has great promise. The advantage of real-time monitoring in food production can further improve the effectiveness of biosensors and encourage their commercial availability and diffusion.

## Figures and Tables

**Figure 1 molecules-26-02940-f001:**
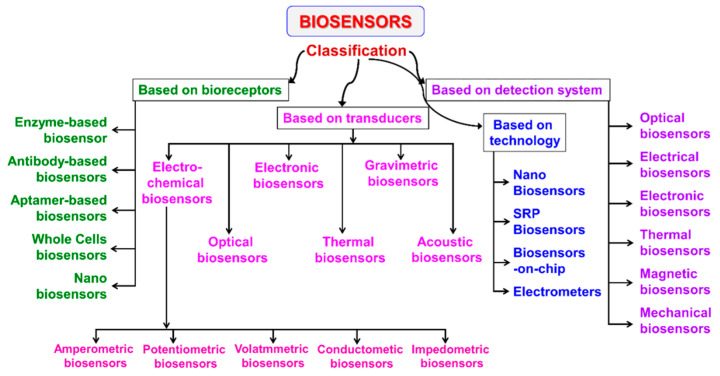
Classification of biosensors based on various bioreceptors and transducers used [16].

**Figure 2 molecules-26-02940-f002:**
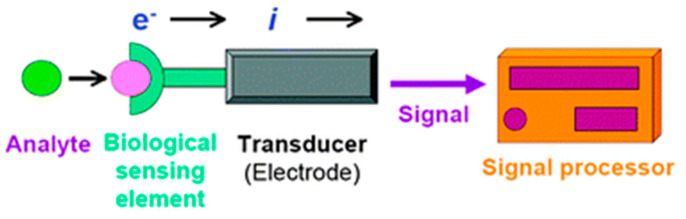
Scheme of a biosensor with an electrochemical transducer. Reprinted with permission from [10] Copyright (2010) Royal Society of Chemistry (RSC).

**Figure 3 molecules-26-02940-f003:**
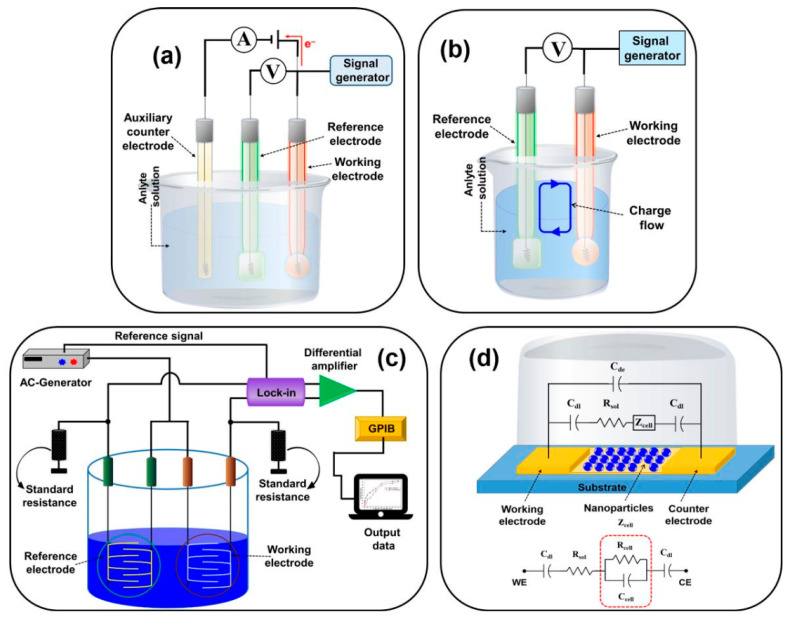
Schematic diagram of (**a**) amperometric/voltammetric, (**b**) potentiometric, (**c**) conductometric biosensors, and (**d**) impedimetric biosensor with the relative equivalent circuit [16] (Cdl = double-layer capacitance of the electrodes, Rsol = resistance of the solution, Cde = capacitance of the electrode, Zcell = impedance introduced by the bound nanoparticles, and Rcell and Ccell are the resistance and capacitance in parallel).

**Figure 4 molecules-26-02940-f004:**
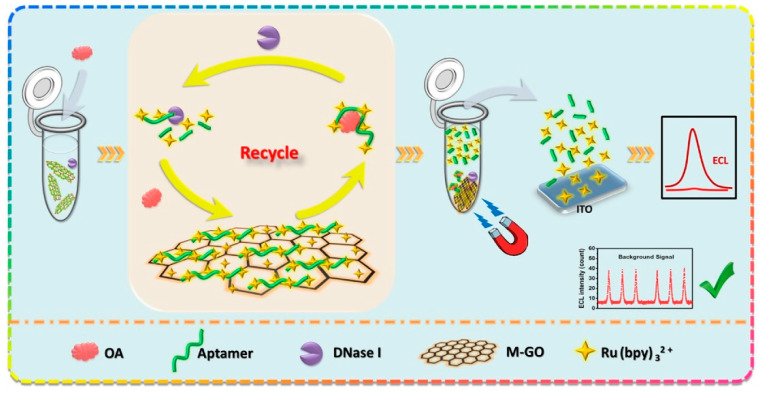
Schematic diagram of the M-GO-assisted homogeneous ECL aptasensor for OA determination. Reprinted with permission from [57] Copyright 2021 Elsevier.

**Figure 5 molecules-26-02940-f005:**
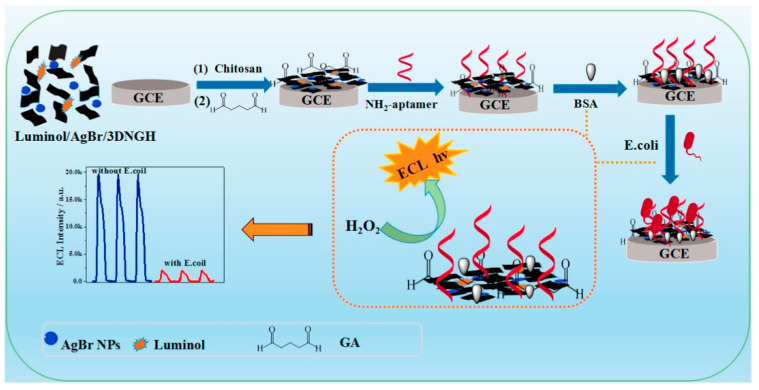
Schematic representation related to the ECL *Escherichia coli* biosensor, fabricated with luminol/AgBr/3DNGH. Reprinted with permission from [116] Copyright 2017 Elsevier.

**Figure 6 molecules-26-02940-f006:**
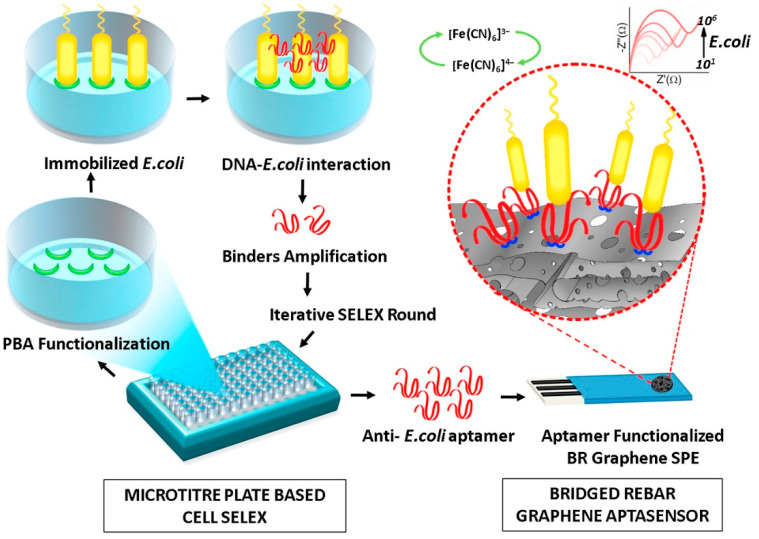
Schematic representation of the assembling and the sensing approach of the *E. coli* aptasensor. Reprinted with permission from [117] Copyright 2017 Elsevier.

**Figure 7 molecules-26-02940-f007:**
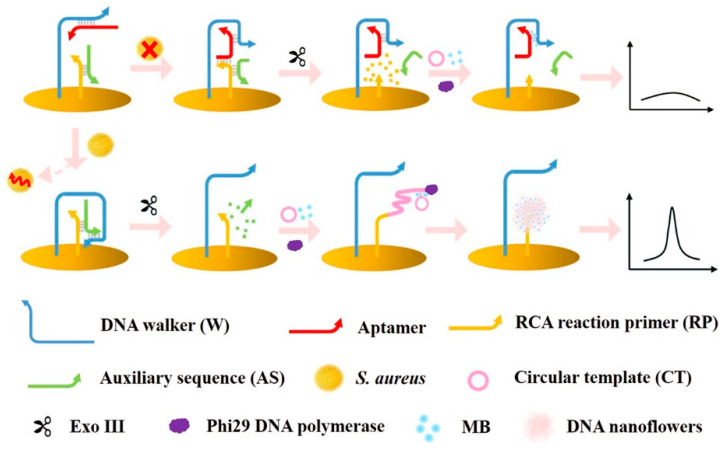
Schematic representation of the biosensor for *S. aureus* based on a DNA walker and DNA nanoflowers. Reprinted with permission from [126] Copyright 2021 American Chemical Society.

**Figure 8 molecules-26-02940-f008:**
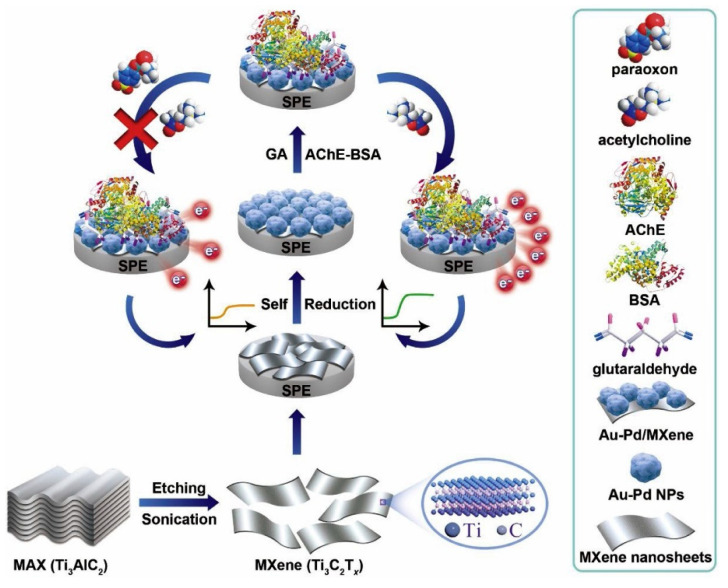
Schematic diagram of the synthesis of MXene nanosheets and assembling of the enzyme-based pesticide biosensor. Reprinted with permission from [150] Copyright 2020 Elsevier.

**Figure 9 molecules-26-02940-f009:**
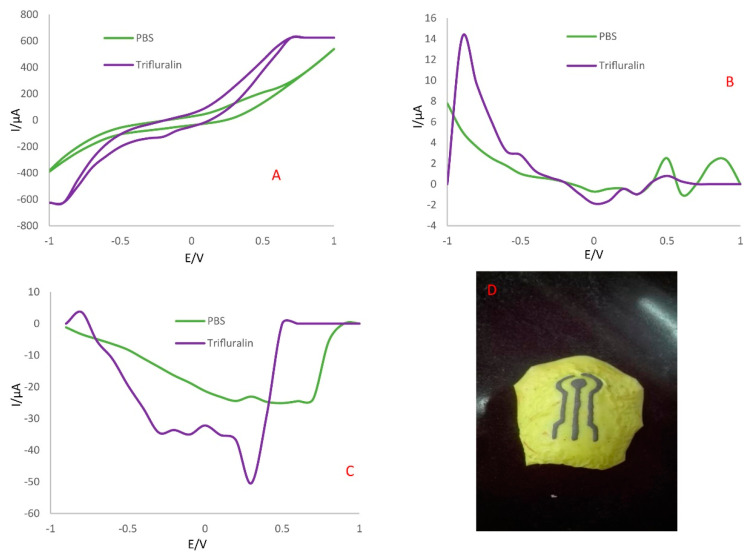
(**A**) Cyclic voltammograms, (**B**) differential pulse voltammograms and (**C**) square wave voltammograms of Ag-citrate/GQDs nano-ink fabricated on the surface of apple-skin incubated at room temperature in the absence and presence of 1 mM trifluralin. Supporting electrolyte is 0.1 M PBS (pH 7.4) in the presence of acetone, (**D**) photographic image of an electrochemical sensor made by direct writing of nano-ink on the surface of apple skin. Reprinted with permission from [158] Copyright 2020 Elsevier.

**Figure 10 molecules-26-02940-f010:**
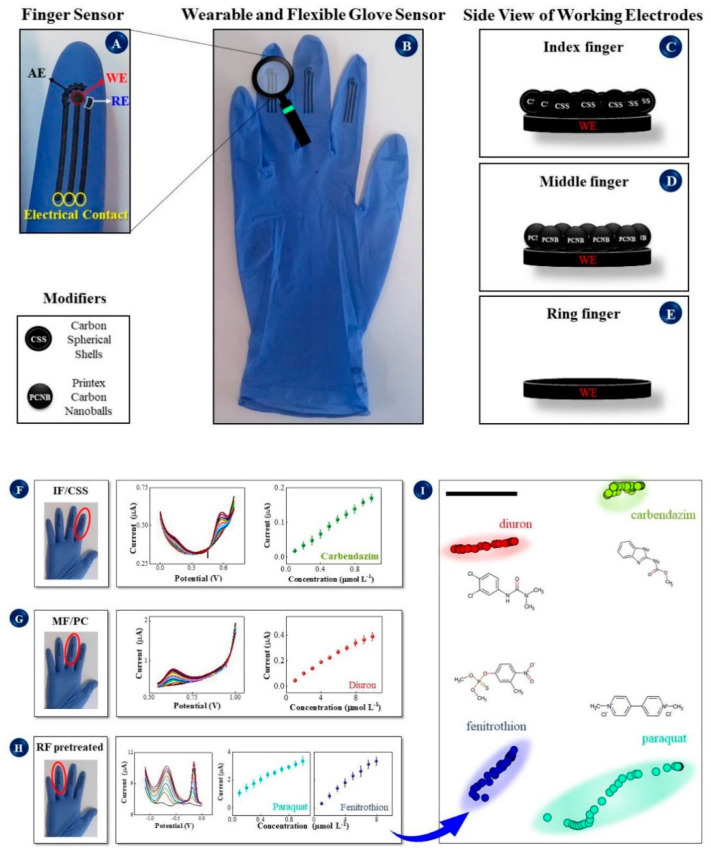
(**A**) Details of finger sensor design with a complete electrochemical system: auxiliary, reference and working electrodes. The connection between electrodes and potentiostat was made via flexible conductive wires for on-site detection. (**B**) Image of the real screen-printed sensing glove. (**C**–**E**) Schematic representation of the side views of CSS, PCNB and pretreated sensing layers for index, middle and ring fingers, respectively. The electrochemical signatures and corresponding analytical curves obtained with index, middle and ring fingers of the glove-embedded sensors are shown in (**F**) through (**H**,**F**) DPV for carbendazim detection from 1.0 × 10^−7^ to 1.0 × 10^−6^ mol L^–1^; (**G**) DPV for diuron detection from 1.0 × 10^−7^ to 1.0 × 10^−6^ mol L^–1^; (**H**) SW voltammograms for paraquat detection from 1.0× from 1.0 × 10^−7^ to 1.0 × 10^−6^ mol L^−1^ and fenitrothion detection from 1.0 × 10^−7^ to 1.0 × 10^−6^ mol L^−1^. Conditions for the detection: 0.1 mol L^−1^ phosphate buffer solution, pH 7.0. (**I**) LSP plot for all pesticides measured with differential pulse and square wave voltammetry, where each voltammogram was converted into a colored dot on the plot. The black bar is only a guide to measure distances between data points. The silhouette coefficient is 0.79. Reprinted with permission from [166] Copyright 2021 Elsevier.

**Figure 11 molecules-26-02940-f011:**
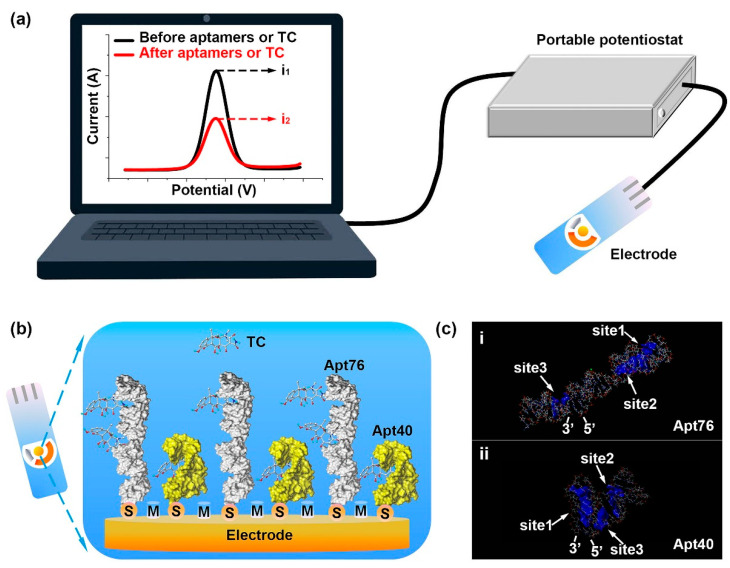
Scheme of the aptamer cocktail-based electrochemical aptasensor. (**a**) The electrochemical sensor was composed of a portable potentiostat, a computer, and an aptamer cocktail functionalized-electrode. (**b**) Working area of the aptamer cocktail-functionalized electrode. Thiolated-Apt76 and thiolated-Apt40 were co-immobilized on the surface of the electrode through S-Au interaction to the capture TC, followed by blocking with mercaptoethanol. S is thiol group; M is ME, 2-mercaptoethanol. (**c**) Predicted binding sites of Apt76 (i) and Apt40 (ii) for TC [178]. Reprinted with permission from [178] Copyright 2019 Elsevier.

**Figure 12 molecules-26-02940-f012:**
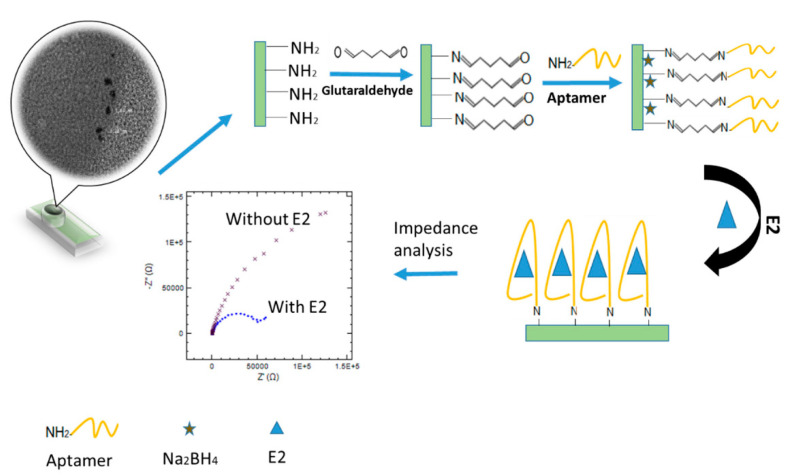
Schematic representation of the immobilization strategy and hybridization detection of 17b-estradiol on aptamer/CDs/SPCE [198].

**Figure 13 molecules-26-02940-f013:**
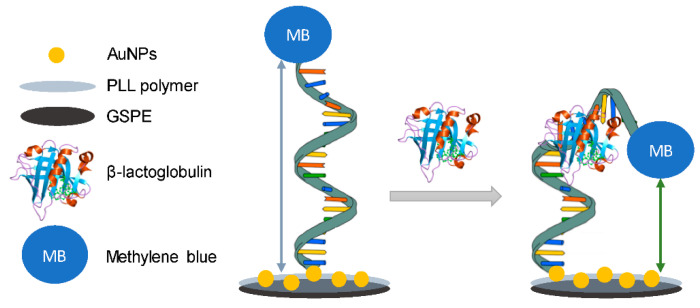
Schematic representation of the sensing strategy of the aptasensor to determine b-lactoglobulin [212].

**Table 4 molecules-26-02940-t004:** An overview of recent electrochemical biosensors for antibiotics determination.

Electrode	(Bio)Sensor Format	Electrochemical Technique	Sample/Analyte	L.R.	LOD	References
GCE	Multiplexed electrochemical aptasensor using metal ions encoded apoferritin probes and double stirring bars-assisted target recycling for signal amplification	SWV	Kanamycin and ampicillin/milk and fish	0.05 pM–50 nM	Kanamycin 18 fM Ampicillin 15 fM	[173]
AuE	Electrochemical aptasensor based on applying a ladder-shaped DNA structure as a multilayer physical block on the surface of a gold electrode	DPV	Ampicillin/milk	7 pM–100 nM	1 pM	[174]
Thin-film gold electrode (TFGE)	Disposable and portable aptasensor using gold nanoparticles (AuNPs)/carboxylated multi-walled carbon nanotubes (cMWCNTs)@thionine connecting complementary strand of aptamer (cDNA) as signal tags	DPV	Oxytetracycline/chicken	1 × 10^−13^–1 × 10^−5^ g mL^−^^1^	3.1 × 10^−14^ g mL^−^^1^	[175]
GCE	Electrochemical aptasensor based on the protective effect of aptamer-antibiotic conjugate towards endonuclease DpnII activity	DPV	Ampicillin/milk and water	0.1–100 nM	32 pM	[176]
AuE	Electrochemical aptasensor incorporating elements of triple-helix aptamer probes (TAP), catalyzed hairpin assembly (CHA) signal amplification and host–guest recognition	DPV	Tetracycline/milk	0.2–100 nM	0.13 nM	[177]
SPAuE	Electrochemical aptasensor based on aptamer cocktail on the surface of gold screen-printed electrodes	DPV	Tetracycline/honey	0.01–1000 ng mL^−1^	0.0073 ng mL^−1^	[178]
AuE	Electrochemical aptasensor based on the classical probe conformation changing mode (PCCM) with a methylene blue (MB) label used as an electrochemical tag	SWV	Kanamycin/milk and water	10.0 nM–10.0 μM	3 nM	[179]
AuE	Electrochemical aptasensor based on the target-induced signal probe shifting (TISPS) method with a free MB label in the assay solution	SWV	Kanamycin/milk and water	200.0 pM–1.0 μM	60 pM	[179]
SPCEs	Potentiometric aptasensor array based on a 4-channel screen-printed carbon electrode	Open-circuit potential (OCP) measurement	Streptomycin and kanamycin/milk	Streptomycin 10 pM–10 μM Kanamicin 10 pM^−^^1^ μM	streptomycin 9.66 pM Kanamycin 5.24 pM	[180]

Abbreviations: AuE: gold electrode; AuNPs: gold nanoparticles; AuSPE: gold screen-printed electrode; cMWCNTs: carboxylated multi-walled carbon nanotubes; CA: chronoamperometry; CF: carbon felt; CHA: catalyzed hairpin assembly; CNF: carbon nanofibers; CV: cyclic voltammetry; DEP: Disposable electrical printed microarray electrode; DPV: differential pulse voltammetry; EIS: electrochemical impedance spectroscopy; GCE: glassy carbon electrode GO: graphene oxide; ITO: indium–tin-oxide electrode: MWCNTs: multi-walled carbon nanotubes; NPG: nanoporous gold; OCP: Open-circuit potential; PGE: pencil graphite electrode; PCCM: probe conformation changing mode; SPCE: screen-printed carbon electrode; SWV: square-wave voltammetry; SWASV: square-wave anodic stripping voltammetry; TAP: triple-helix aptamer probe; TISPS: target-induced signal probe shifting; TFGE: thin-film gold electrode.

## Data Availability

No new data were created or analyzed in this study. Data sharing does not apply to this article.

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
