# Peer review of "Electrochemical Biosensors in Food Safety: Challenges and Perspectives"

_molecules, 2021, doi:10.3390/molecules26102940_

Round 1
Reviewer 1 Report
Antonella et al. reviewed the recent developments in food safety detection by electrochemical biosensors and overall introduced the application of electrochemical biosensors in food screening analysis. This review focused on the detection of some of the most important bacteria, toxins, pesticides, antibiotics, and contaminants in foods. This work is beneficial to develop more performant detection-technology in food safety. While the text seems generally comprehend, there are few crucial points must be addressed one by one.
- If the author wants to exhibit the advantages of electrochemical biosensors in food safety detections relative to traditional detection devices, more details about electrochemical biosensors, such as the comparison between of different principles, sensing materials, device structure, and some of presentative work figures, should be introduced instead of only electrochemical techniques and electrode materials.
- Although the overview of recent electrochemical biosensors for different important indexes in others’ works was presented in this review, a brief conclusion or the deep summary is still missing and the author did not clarify clear difference among similar works.
- Biosensors can significantly support the screening of food chain and products owing to its low-cost, fast analyses, and not need skilled personal, but many nano or micro chemical material is poisonous or biologically incompatible, so that the secondary pollution possibly lead to these methods not suitable for foods screening. Hence, the author should analyze the secondary pollutions from different electrode materials to foods.
Author Response
Please the attachment

Reviewer 2 Report
The review manuscript entitled « Electrochemical biosensors in food safety : challenges and perspectives » is composed of 47 pages (without references), 6 tables, 8 figures and 213 references.
The review starts with an introduction about the interest for the conception of electrochemical sensors in the field of food safety, then the basics of electrochemical sensors and electrode materials. The main part of the manuscript (42 pages) presents the latest articles concerning the detection of toxins, pathogenic bacteria, pesticides, antibiotics, endocrine-disrupting chemicals and allergens.
Several specialized and recent reviews concerning specific targets are already available and cited, so that the proposed review focuses on the more recent approaches for a given target.
The title of the review is “Electrochemical biosensors in food safety : challenges and perspectives “ but, to my opinion, the critical discussion with the challenges and perspectives about the proposed approaches in the literature for a given target should be developed : for example, intermediate conclusions could be added at the end of each section (3.1 to 3.6), including current remaining challenges and perspectives.
In each section, sub-sections could be added, dealing with a given target, in order to facilitate reading. In section 3.1, it could be interesting to indicate in the text at what concentration the target is dangerous for humans and/or animals. More illustrations (figures) could be given, by composing one figure with different schemes from the literature for example. I also would suggest to group together the paragraphs concerning articles using the same type of detection (either EIS, evening DPV etc) or same type of bioreceptor. Finally, there is little discussion of the multi-detection aspect; this could be discussed in part 4 (conclusions).
The author should also address the following points:
- The description of an approach (of an article) should be done in one paragraph (without line breaks) to simplify the reading.
- In each table, for a given target, please use the same unit of LOD. Also, at the end of each table, the abbrevations used should be explained.
- When a review is cited in the test, it could be interesting to add in the text the year of the review in all cases.
- Lines 425-426: The sentence “The sensor avoids the direct contact between the reaction system and the signal measurement system” is not clear; please reformulate.
- At the end of line 453, the author refers to Elisa test : can the author indicate the performances of Elisa test for AFB1 detection
- Paragraph lines 577-581: The orientated and non-orientated immobilizations of antibodies should be briefly explained.
- Lines 697-698: “The aptamer/target interaction on the conjugated copolymer and the copolymer conductivity improved the impedimetric measurements.” Can the author gives more details about the interactions and the conductivity and why this improves the measurements
- Lines 739-740 « UHT whole milk samples were artificially spiked, and the obtained results are very promising » : can the author comment on how « the results are very promising »
- Line 795: the author should indicate how the complementary DNA was immobilized on the electrode (SAM?)
- The explanations concerning the operation several biosensors should be included for:
- The sentence “An electrochemical biosensor for rapid detection of S. aureus based on silver wire across electrodes was reported by He” for reference 115 needs explanation.
- The sentence “An electrochemical biosensor for Staphylococcus aureus was designed, based on a triple-helix molecular switch, which can control the switching of electrochemical signals “ from reference 116: more details should be given to better understand the principle of detection
- Lines 1138-1139: Please add explanation abou the principle of the measurements from the reference 145.
- Line 1238: Please explain how the nanoporous gold presents the properties of recognition
- Line 1190 : please add a description and comment concerning Figure 6
- Line 1266 : The caption of Figure 7 should be detailed
- Line 1300 : can the author add the main values of MRL for common antibiotics
- Line 1344 : The sentence « Another aptasensor, based on the protective effect of aptamer-antibiotic conjugate towards endonuclease DpnII activity, was developed [167] » : the principle is not clear, can the author reformulate the sentence
- Line1369 : The caption of Figure 8 should be detailed
- One illustration (figure) coud be given for Part 3.5 and Part 3.6
- Lines 1462-1466 : can the author include the principle of detection used in refernce 178
- Line 1755 : the nature of the MIPs (reference 123) should be indicated
- Some sentences are the same as in the abstracts of the cited articles (see below): Please cite or modify these sentences
- Lines 305-306 “The composite shows a ratiometric response in the UV-Vis absorption spectrum and quenching in the fluorescence profile with a detection limit of 20 nM for OA in aqueous medium” : this sentence is the same as in the abstract of the cited article 53.
- Lines 995-996: “The sensor is incorporated into a particle/sediment trap for the real-time analysis of irrigation water in a hydroponic lettuce system. » : this sentence is the same as in the abstract of the cited article 129.
- Lines 1004-1005 “the development of an electrochemical immunosensor for rapid, specific, and decentralized detection of the invasion-associated protein p60 secreted by Listeria monocytogenes” : this sentence is the same as in the abstract of the cited article 129.
- Minor remarks
- along the text “in situ” should be in italic.
- Lines 282-285: the sentence contains no verb.
- Lines 994-995 : this sentence is missing a verb.
